# ARMA Nets:
# Expanding Receptive Field for Dense Prediction

**Jiahao Su**[1]
jiahaosu@umd.edu

**Shiqi Wang**[2]
161170041@smail.nju.edu.cn

**Furong Huang**[1]
furongh@cs.umd.edu

[1]University of Maryland, College Park, MD USA   [2]Nanjing University, Nanjing, China

## Abstract

Global information is essential for dense prediction problems, whose goal is to compute a discrete or continuous label for each pixel in the images. Traditional convolutional layers in neural networks, initially designed for image classification, are restrictive in these problems since the filter size limits their receptive fields. In this work, we propose to replace any traditional convolutional layer with an autoregressive moving-average (ARMA) layer, a novel module with an adjustable receptive field controlled by the learnable autoregressive coefficients. Compared with traditional convolutional layers, our ARMA layer enables explicit interconnections of the output neurons and learns its receptive field by adapting the autoregressive coefficients of the interconnections. ARMA layer is adjustable to different types of tasks: for tasks where global information is crucial, it is capable of learning relatively large autoregressive coefficients to allow for an output neuron's receptive field covering the entire input; for tasks where only local information is required, it can learn small or near zero autoregressive coefficients and automatically reduces to a traditional convolutional layer. We show both theoretically and empirically that the effective receptive field of networks with ARMA layers (named ARMA networks) expands with larger autoregressive coefficients. We also provably solve the instability problem of learning and prediction in the ARMA layer through a re-parameterization mechanism. Additionally, we demonstrate that ARMA networks substantially improve their baselines on challenging dense prediction tasks, including video prediction and semantic segmentation. Our code is available on `https://github.com/umd-huang-lab/ARMA-Networks`.

## 1 Introduction

Convolutional layers in neural networks have many successful applications for machine learning tasks. Each output neuron encodes an input region of the network measured by the *effective receptive field* (ERF) [25]. A large ERF that allows for sufficient global information is needed to make accurate predictions; however, a simple stack of convolutional layers does not effectively expand ERF. Convolutional neural networks (CNNs) typically encode global information by adding downsampling (pooling) layers, which coarsely aggregate global information. A fully-connected classification layer subsequently reduces the entire feature map to an output label. Downsampling and fully-connected layers are suitable for image classification tasks where only a single prediction is needed. But they are less effective, due to potential loss of information, in dense prediction tasks such as semantic segmentation and video prediction, where each pixel requests a prediction. Therefore, it is crucial to introduce mechanisms that enlarge ERF without too much information loss.

Naive approaches to expanding ERF, such as deepening the network or enlarging the filter size, drastically increase the model complexity, which results in expensive computation, difficulty in optimization, and susceptibility to overfitting. Advanced architectures have been proposed to expand ERF, including encoder-decoder networks [30], dilated convolutional networks [40, 41], and non-

local networks [34]. However, encoder-decoder networks could lose high-frequency information due to the downsampling layers. Dilated convolutional networks could suffer from the gridding effect while the ERF expansion is limited, and non-local networks are expensive in training and inference.

We introduce a novel *autoregressive-moving-average* (ARMA) layer that enables adaptive receptive field by explicit interconnections among its output neurons. Our ARMA layer realizes these interconnections via extra convolutions on output neurons, on top of the convolutions on input neurons as in a traditional convolutional layer. We provably show that an ARMA network can have arbitrarily large ERF, thus capturing global information, with minimal extra parameters at each layer. Consequently, an ARMA network can flexibly enlarge its ERF to leverage global knowledge without reducing spatial resolution. Moreover, the ARMA networks are independent of the architectures above, including encoder-decoder networks, dilated convolutional networks, and non-local networks.

A significant challenge in ARMA networks lies in the complex computations needed in both forward and backward propagations — simple convolution operations are not applicable since the output neurons are influenced by their neighbors and thus interrelated. Another challenge in ARMA networks is instability — the additional interconnections among the output neurons could recursively amplify the outputs and lead them to infinity. We address both challenges in this paper.

**Summary of Contributions**

- We introduce a novel ARMA layer that is a plug-and-play module substituting convolution layers in neural networks to allow flexible tuning of their ERF, adapting to the task requirements, and improving performance in dense prediction problems.

- We recognize and address the problems of *computation* and *instability* in ARMA layers. **(1)** To reduce computational complexity, we develop FFT-based algorithms for both forward and backward passes; **(2)** To guarantee stable learning and prediction, we propose a *separable ARMA layer* and a re-parameterization mechanism that ensures the layer to operate in a stable region.

- We successfully apply ARMA layers in ConvLSTM network [39] for pixel-level multi-frame video prediction and U-Net model [30] for medical image segmentation. ARMA networks substantially outperform the corresponding baselines on both tasks, suggesting that our proposed ARMA layer is a general and useful building block for dense prediction problems.

## 2  Related Works

**Dilated convolution [15]** enlarges the receptive field by upsampling the filter coefficients with zeros. Unlike encoder-decoder structure, dilated convolution preserves the spatial resolution and is thus widely used in dense prediction problems, including semantic segmentation [7, 24, 40], and objection detection [10, 20]. However, dilated convolution by itself creates gridding artifacts if its input contains higher frequency than the upsampling rate [41], and the inconsistency of local information hampers the performance of the dilated convolutional networks [35]. Such artifacts can be alleviated by extra anti-aliasing layer [41], group interacting layer [35] or spatial pyramid pooling [8].

**Deformable convolution** allows the filter shape (i.e., locations of the incoming pixels) to be learnable [11, 16, 42]. While deformable convolution focuses on adjusting the filter *shape*, our ARMA layer aims to expand the effective filter *size* adaptively.

**Non-local attention network [34]** inserts non-local attention blocks between the convolutional layers. A non-local attention block computes a weighted sum of all input neurons for each output neuron, similar to attention mechanism [33]. In practice, non-local attention blocks are computationally expensive, thus they are typically inserted in the upper part of the network (with lower resolution). In contrast, our ARMA layers are economical (see section 4) and can be used throughout the network.

**Encoder-decoder structured network** pairs each downsampling layer with another upsampling layer to maintain the resolution, and introduces skip-connection between the pair to preserve the high-frequency information [24, 30]. Since the shortcut bypasses the downsampling/upsampling layers, the network has a small receptive field for the high-frequency components. A potential solution is to augment upsampling with non-local attention block [27] or ARMA layer (section 6).

**Recurrent neural networks** over the spatial domain [5, 17, 22, 23, 28, 32] are used to expand the receptive field or learn the affinity between neighboring pixels. However, almost all prior works consider nonlinear recurrent neural networks, where the activation in recursions prohibits an efficient

parallel algorithm. Quasi-recurrent neural networks [4] partially address the problem by decoupling the linear operations and parallelizing them using convolutions. In contrast, our proposed ARMA layer is equivalent to a linear recurrent neural network, allowing for efficient evaluation using FFT.

# 3 ARMA Neural Networks

In this section, we introduce a novel *autoregressive-moving-average* (ARMA) layer and analyze its ability to expand *Effective Receptive Field* (ERF) in neural networks. The analysis is further verified by visualizing the ERF with varying network depth and strength of autoregressive coefficients.

## 3.1 ARMA Layer

A traditional convolutional layer is essentially a *moving-average* model [3]:

$$\mathcal{Y}_{:,:,t} = \sum_{s=1}^{S} \mathcal{W}_{:,:,t,s} * \mathcal{X}_{:,:,s}, \tag{1}$$

where the *moving-average coefficients* $\mathcal{W} \in \mathbb{R}^{K_m \times K_m \times T \times S}$, is parameterized by a 4th-order kernel ($K_m$ is the filter size, and $S, T$ are input/output channels), : denotes all elements from the specified coordinate and $*$ denotes convolution between an input feature and a filter.

As motivated in the introduction, we introduce a novel ARMA layer that enables an adaptive receptive field by introducing explicit interconnections among its output neurons, as illustrated in Figure 1. Our ARMA layer realizes these interconnections by introducing extra convolutions on the outputs, upon the convolutions on the inputs as in a traditional convolutional layer. As a result, in an ARMA layer, each output neuron can be affected by an input pixel faraway through interconnections among the output neurons, thus receives global information. Formally, we define an ARMA layer in Definition 1.

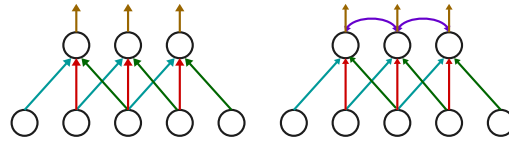

**(a)** Convolution      **(b)** ARMA

**Figure 1:** The ARMA layer introduces interconnections among output neurons explicitly.

**Definition 1** (**ARMA Layer**). *An ARMA layer is parameterized by a moving-average kernel (coefficients) $\mathcal{W} \in \mathbb{R}^{K_m \times K_m \times S \times T}$ and an autoregressive kernel (coefficients) $\mathcal{A} \in \mathbb{R}^{K_a \times K_a \times T}$. It receives an input $\mathcal{X} \in \mathbb{R}^{I_1 \times I_2 \times S}$ and returns an output $\mathcal{Y} \in \mathbb{R}^{I_1' \times I_2' \times T}$ with an ARMA model:*

$$\mathcal{A}_{:,:,t} * \mathcal{Y}_{:,:,t} = \sum_{s=1}^{S} \mathcal{W}_{:,:,t,s} * \mathcal{X}_{:,:,s}. \tag{2}$$

*Remarks:* **(1)** The ARMA layer maintains the *shift-invariant* property, since the output interconnections are realized by convolutions. **(2)** The ARMA layer *reduces* to a traditional layer if the autoregressive kernel $\mathcal{A}$ represents an identical mapping. **(3)** The ARMA layer is a *plug-and-play* module that can replace *any* convolutional layer, adding $K_a^2 T$ extra parameters negligible compared to $K_w^2 ST$ parameters in a traditional convolution layer. **(4)** Unlike a traditional layer, computing Equation 2 and its backpropagation are nontrivial, studied in section 4.

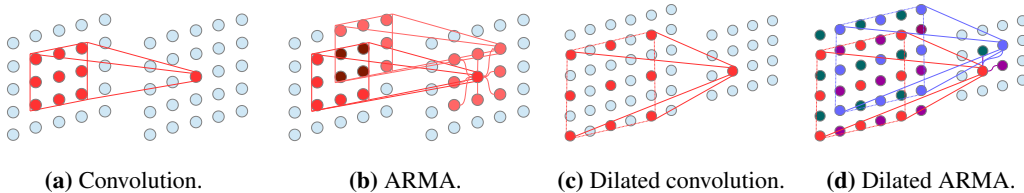

**(a)** Convolution.      **(b)** ARMA.      **(c)** Dilated convolution.      **(d)** Dilated ARMA.

**Figure 2: Diagrams of receptive field.** In **(b)**, each output neuron receives its neighbors' receptive field. In **(d)**, ARMA's autoregression fills the gaps created by dilated convolution.

Our ARMA layer can be combined with the methods of dilated convolutional layer, deformable convolutional layer, non-local attention block, as well as encoder-decoder architecture. For instance, a *dilated ARMA layer*, illustrated in Figure 2d, removes the gridding effect caused by dilated convolution — the autoregressive kernel is interpreted as an anti-aliasing filter.

The motivation of introducing the ARMA layer is to enlarge the effective input region for each network output without increasing the filter size or network depth, thus avoiding the difficulties in training larger or deeper models. As illustrated in Figure 2, each output neuron in a traditional convolutional layer (Figure 2a) only receives information from a small input region (the filter size). However, an ARMA layer enlarges the small local region to a larger one (Figure 2b). It enables an output neuron to receive information from a faraway input neuron through the connections to its neighbors. In subsection 3.2, we formally introduce the concept of *effective receptive field* (ERF) to characterize the input region size. Moreover, we will show that an ARMA network can have arbitrarily large ERF with a single extra parameter at each layer in Theorem 3.

## 3.2 Effective Receptive Field

Effective receptive field (ERF) [25] measures the area of the input region that makes *substantial* contribution to an output neuron. In this section, we analyze the ERF size of an $L$-layers network with ARMA layers v.s. traditional convolutional layers. Formally, consider an output at location $(i_1, i_2)$, the impact from an input pixel at $(i_1 - p_1, i_2 - p_2)$ (i.e $L$ layers and $(p_1, p_2)$ pixels away) is measured by the amplitude of partial derivative $g(i_1, i_2, p_1, p_2) = \left| \partial \mathcal{Y}_{i_1,i_2,t}^{(L)} / \partial \mathcal{X}_{i_1-p_1,i_2-p_2,s}^{(1)} \right|$ (where superscripts index the layers), i.e., how much the output changes as the input pixel is perturbed.

**Definition 2** (**Effective Receptive Field, ERF**). *Consider an $L$-layers network with an $S$-channels input $\mathcal{X}^{(1)} \in \mathbb{R}^{I_1 \times I_2 \times S}$ and a $T$-channels output $\mathcal{Y}^{(L)} \in \mathbb{R}^{I_1 \times I_2 \times T}$, its effective receptive field is defined as the empirical distribution of the gradient maps: $ERF(p_1, p_2) = 1/(I_1 I_2 S T) \cdot \sum_{s,t,i_1,i_2} [g(i_1, i_2, p_1, p_2) / \sum_{j_1,j_2} g(j_1, j_2, p_1, p_2)]$, To measure the size of the ERF, we define its radius $r(ERF)$ as the standard deviation of the empirical distribution:*

$$r(ERF)^2 = \sum_{p_1, p_2} (p_1^2 + p_2^2) \, ERF(p_1, p_2) - \left[ \sum_{p_1, p_2} \sqrt{p_1^2 + p_2^2} \, ERF(p_1, p_2) \right]^2. \qquad (3)$$

Notice that ERF simultaneously depends on the model parameters and a specified input to the network, i.e., ERF is both *model-dependent* and *data-dependent*. Therefore, it is generally intractable to compute the ERF analytically for any practical neural network.

We follow the original paper of ERF [25] to estimate the radius with a simplified linear network. The paper empirically verifies that such an estimation is accurate and can be used to guide filter designs.

**Theorem 3** (**ERF of a Linear ARMA Network**). *Consider an $L$-layers linear network, where the $\ell^{th}$ layer computes $y_i^{(\ell)} - a^{(\ell)} y_{i-1}^{(\ell)} = \sum_{p=0}^{K^{(\ell)}-1} [(1 - a^{(\ell)})/K^{(\ell)}] \cdot y_{i-d^{(\ell)}p}^{(\ell-1)}$ (i.e., the moving-average coefficients are uniform with length $K^{(\ell)}$ and dilation $d^{(\ell)}$, and the autoregressive coefficients $\boldsymbol{a}^{(\ell)} = \{1, -a^{(\ell)}\}$ has length 2 ). Suppose $0 \le a^{(\ell)} < 1, \forall \ell \in [L]$, the ERF radius of the network is*

$$r(ERF)_{ARMA}^2 = \sum_{\ell=1}^{L} \left[ \frac{d^{(\ell)2} \left( K^{(\ell)2} - 1 \right)}{12} + \frac{a^{(\ell)}}{\left(1 - a^{(\ell)}\right)^2} \right]. \qquad (4)$$

We prove Theorem 3 in Appendix B. If the coefficients for different layers are identical, e.g., $K^{(\ell)} = K, d^{(\ell)} = d, a^{(\ell)} = a$, the radius reduces to $r(ERF)_{ARMA} = \sqrt{L} \cdot \sqrt{d^2(K^2 - 1)/12 + a/(1 - a)^2}$. Moreover, if $a = 0$ and $d = 1$, the ARMA layers reduce to traditional convolutional layers, and the ERF of the resulted linear CNN has radius $r(ERF)_{CNN} = \sqrt{L} \cdot \sqrt{(K^2 - 1)/12}$ as shown in [25].

*Remarks:* **(1) Compared with a (dilated) CNN, an ARMA network can have arbitrarily large ERF with an extra parameter $a$ at each layer.** When the autoregressive coefficient $a$ is large (e.g., $a > 1 - 1/(dK)$), the second term $a/(1 - a)^2$ dominates the radius, and the ERF is substantially larger than that of a CNN. In particular, the radius tends to infinity as $a$ approaches 1. **(2) An ARMA network can adaptively adjust its ERF through learnable parameter $a$.** As $a$ gets smaller (e.g., $a < 1 - 1/(dK)$), the second term is comparable to or smaller than the first term, and the effect of expanded ERF diminishes. In particular, if $a = 0$, an ARMA network reduces to a CNN.

**Visualization of the ERF.** We analytically show in Theorem 3 that the ERF radius increases with the network depth and autoregressive coefficients' magnitude. We now verify our analysis by simulating linear ARMA networks with varying depths and autoregressive coefficients' magnitude. As shown in Figure 3, the ERF radius increases as the autoregressive coefficients get larger. When the autoregressive coefficients are zeros, an ARMA network reduces to a traditional convolutional network. The simulation results also indicate that ARMA's ability to expand the ERF increases with the network depth. In conclusion, an ARMA network can have a large ERF even when the network is shallow, and its ability to expand the ERF increases when the network gets deeper.

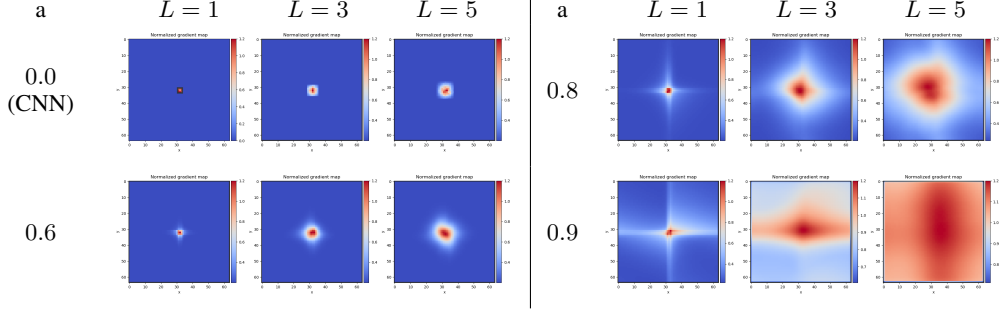

**Figure 3:** Visualization of ERF in linear ARMA networks under different network depth $L = 1, 3, 5$ and different magnitude of the autoregressive coefficient $a = 0.0, 0.6, 0.8, 0.9$ (See subsection A.1 for detailed experimental setup.)

## 4 Prediction and Learning of ARMA Layers

In an ARMA layer, each neuron is influenced by its neighbors from all directions (see Figure 2b). As a result, no neuron could be evaluated alone before evaluating any other neighboring neurons. To compute Equation 2, we need to solve a set of linear equations to obtain all values simultaneously. **(1)** However, the standard solver using Gaussian elimination is too expensive to be practical, and therefore we need to seek a more efficient solution. **(2)** Furthermore, the solver for linear equations typically does not support automatic differetiation, and we have to derive the backward equations analytically. **(3)** Finally, we also need to devise an efficient algorithm to compute the backpropagation equations efficiently. In the section, we address these aforementioned problems.

**Decomposing an ARMA Layer.** We decompose the ARMA layer in Equation 2 into a moving-average (MA) layer and an autoregressive (AR) layer:

$$\text{MA Layer: } \mathcal{T}_{:,:,t} = \sum_{s=1}^{S} \mathcal{W}_{:,:,t,s} * \mathcal{X}_{:,:,s}; \tag{5a}$$

$$\text{AR Layer: } \mathcal{A}_{:,:,t} * \mathcal{Y}_{:,:,t} = \mathcal{T}_{:,:,t}, \tag{5b}$$

where $\mathcal{T} \in \mathbb{R}^{I_1' \times I_2' \times T}$ is the intermediate result.

**Difficulty in Computing the AR Layer.** While the MA layer in Equation 5a is simply a traditional convolutional layer (Equation 1), it is nontrivial to solve the AR layer in Equation 5b. Naively using Gaussian elimination, the linear equations in the AR layer can be solved in time cubic in dimension $O((I_1^2 + I_2^2)I_1 I_2 T)$, which is too expensive.

**Solving the AR Layer.** We propose to use the frequency-domain division [21] to solve the *deconvolution* problem in the AR layer. Since the convolution in the spatial domain leads to an element-wise product in the frequency domain, we first transform $\mathcal{A}, \mathcal{T}$ into their frequency representa-

| Layer | # params. | # FLOPs | $r(ERF)^2$ |
|---|---|---|---|
| Conv. | $K_w^2 C^2$ | $O(I^2 K_w^2 C^2)$ | $O(L K_w^2)$ |
| ARMA | $K_w^2 C^2$ $+ K_a^2 C$ | $O(K_w^2 I^2 C^2 +$ $I^2 \log(I)\, C)$ | $O\big(L K_w^2 +$ $L \dfrac{a}{(1-a)^2}\big)$ |

**Table 1:** An ARMA layer achieves large gain of the ERF radius through small overhead of extra # of parameters and # of FLOPs. Through a single extra parameter $a$ (thus $K_a = 2$), the ERF radius can be arbitrarily large. For notational simplicity, we assume all heights and widths are equal $I_1 = I_2 = I_1' = I_2' = I$, and the input and output channels are the same $S = T = C$.

tions $\widetilde{\mathcal{A}}, \widetilde{\mathcal{T}}$, with which we compute $\widetilde{\mathcal{Y}}$ (the frequency representation of $\mathcal{Y}$) with the element-wise division. Then, we reconstruct the output $\mathcal{Y}$ by an inverse Fourier transform of $\widetilde{\mathcal{Y}}$.

**Computational Overhead.** ARMA trades small overhead of extra parameters and computation for a large gain of ERF radius, as shown in Table 1. With *Fast Fourier Transform* (FFT), the FLOPS required by the extra autoregressive layer is $O(\log(\max(I_1, I_2))I_1 I_2 T)$ (see Appendix C for derivations). Importantly, compared with non-local attention block [34], the extra computation introduced in an ARMA layer is smaller; a non-local attention block requires $O(I_1^2 I_2^2 T)$ FLOPS.

**Backpropagation.** Deriving the backpropagation for an ARMA layer is nontrivial; although the backpropagation rule for the MA layer is conventional, that of the AR layer is not. In Theorem 4, we show that the backpropagation of an AR layer can be computed as two ARMA models.

**Theorem 4 (Backpropagation of an AR layer).** *Given $\mathcal{A}_{:,:,t} * \mathcal{Y}_{:,:,t} = \mathcal{T}_{:,:,t}$ and the gradient $\partial\mathcal{L}/\partial\mathcal{Y}$, the gradients $\{\partial\mathcal{L}/\partial\mathcal{A}, \partial\mathcal{L}/\partial\mathcal{X}\}$ can be obtained by two ARMA models:*

$$\mathcal{A}_{:,:,t}^{\top} * \frac{\partial\mathcal{L}}{\partial\mathcal{A}_{:,:,t}} = -\mathcal{Y}_{:,:,t}^{\top} * \frac{\partial\mathcal{L}}{\partial\mathcal{Y}_{:,:,t}}; \tag{6a}$$

$$\mathcal{A}_{:,:,t}^{\top} * \frac{\partial\mathcal{L}}{\partial\mathcal{T}_{:,:,t}} = \frac{\partial\mathcal{L}}{\partial\mathcal{Y}_{:,:,t}}. \tag{6b}$$

*where $\mathcal{A}_{:,:,t}^{\top}$ and $\mathcal{Y}_{:,:,t}^{\top}$ are the transposed images of $\mathcal{A}_{:,:,t}$ and $\mathcal{Y}_{:,:,t}$ (e.g., $\mathcal{A}_{i_1,i_2,t}^{\top} = \mathcal{A}_{-i_1,-i_2,t}$).*

Since the backpropagation is characterized by ARMA models, it can be evaluated efficiently using FFT similar to Equation 5. The proof of Theorem 4, with its FFT evaluation, is given in Appendix C.

## 5 Stability of ARMA Layers

An ARMA model with arbitrary coefficients is not always stable. For example, the model $y_i - ay_{i-1} = x_i$ is unstable if $|a| > 1$: Consider an input $\boldsymbol{x}$ with $x_0 = 1$ and $x_i = 0, \forall i \neq 0$, the output $\boldsymbol{y}$ will recursively amplify itself as $y_0 = 1, y_1 = a, \cdots, y_i = a^i$ and diverge to infinity.

### 5.1 Stability Constraints for an ARMA Layer

The key to guaranteeing the stability of an ARMA layer is to constrain its autoregressive coefficients, which prevents the output from repeatedly amplifying itself. To derive the constraints, we propose a special design, *separable ARMA layer* inspired by *separable filters* [21].

**Definition 5 (Separable ARMA Layer).** *A separable ARMA layer is parameterized by a moving-average kernel $\mathcal{W} \in \mathbb{R}^{K_w \times K_w \times S \times T}$ and $T \times Q$ sets of autoregressive filters $\{(f_{:,t}^{(q)}, g_{:,t}^{(q)})_{q=1}^{Q}\}_{t=1}^{T}$. It takes an input $\mathcal{X} \in \mathbb{R}^{I_1 \times I_2 \times S}$ and returns an output $\mathcal{Y} \in \mathbb{R}^{I_1' \times I_2' \times T}$ as*

$$\left( f_{:,t}^{(1)} * \cdots * f_{:,t}^{(Q)} \right) \otimes \left( g_{:,t}^{(1)} * \cdots * g_{:,t}^{(Q)} \right) * \mathcal{Y}_{:,:,t} = \sum_{s=1}^{S} \mathcal{W}_{:,:,t,s} * \mathcal{X}_{:,:,s} \tag{7}$$

*where the filters $f_{:,t}^{(q)}, g_{:,t}^{(q)} \in \mathbb{R}^3$ are length-3, and $\otimes$ denotes outer product of two 1D-filters.*

*Remarks:* Each autoregressive filter $\mathcal{A}_{:,:,t}$ is designed to be separable, i.e., $\mathcal{A}_{:,:,t} = F_{:,t} \otimes G_{:,t}$, thus it can be characterized by 1D-filters $F_{:,t}$ and $G_{:,t}$. By the fundamental theorem of algebra [29], any 1D-filter can be represented as a composition of length-3 filters. Therefore, $F_{:,t}$ and $G_{:,t}$ can further be factorized as $F_{:,t} = f_{:,t}^{(1)} * f_{:,t}^{(2)} \cdots * f_{:,t}^{(Q)}$ and $G_{:,t} = g_{:,t}^{(1)} * g_{:,t}^{(2)} \cdots * g_{:,t}^{(Q)}$. In summary, each $\mathcal{A}_{:,:,t}$ is characterized by $Q$ sets of length-3 autoregressive filters $(f_{:,t}^{(q)}, g_{:,t}^{(q)})_{q=1}^{Q}$.

**Theorem 6 (Constraints for a Stable Separable ARMA Layer).** *A sufficient condition for the separable ARMA layer (Definition 5) to be stable (i.e., output be bounded for any bounded input) is:*

$$\left| f_{-1,t}^{(q)} + f_{1,t}^{(q)} \right| < f_{0,t}^{(q)}, \ \left| g_{-1,t}^{(q)} + g_{1,t}^{(q)} \right| < g_{0,t}^{(q)}, \ \forall q \in [Q], t \in [T]. \tag{8}$$

The proof is deferred to Appendix D, which follows the standard techniques using Z-transform.

### 5.2 Achieving Stability via Re-parameterization

In principle, the constraints required for stability in an ARMA layer (as in Theorem 6) could be enforced through constraints in optimization. However, a constrained optimization algorithm, such as

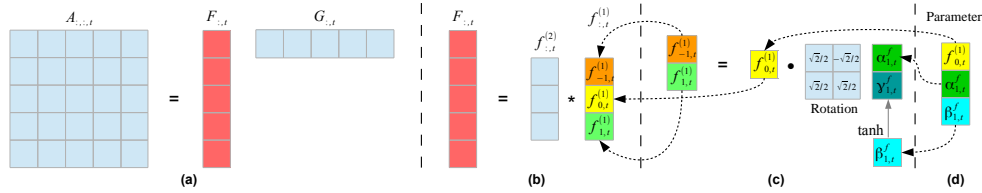

**Figure 4:** For each channel $t$, **(a)** the two-dimensional filter $\mathcal{A}_{:,:,t}$ is parameterized through an outer product of two 1D-filters $F_{:,t}$ and $G_{:,t}$; **(b)** $F_{:,t}$ is parameterized through a convolution of $f_{:,t}^{(1)} * \cdots * f_{:,t}^{(Q)}$, and similarly $G_{:,t}$ as a convolution of $g_{:,t}^{(1)} * \cdots * g_{:,t}^{(Q)}$; **(c)** we re-parameterize each constrained $(f_{-1,t}^{(q)}, f_{1,t}^{(q)})$ to unconstrained $(\alpha_{q,t}^f, \beta_{q,t}^f)$, and similarly $(g_{-1,t}^{(q)}, g_{1,t}^{(q)})$ to $(\alpha_{q,t}^g, \beta_{q,t}^g)$; **(d)** final parameters for unconstrained optimization are $(f_{0,t}^{(q)}, \alpha_{q,t}^f, \beta_{q,t}^f, g_{0,t}^{(q)} \alpha_{q,t}^g, \beta_{q,t}^g)_{q=1}^Q$.

projected gradient descent [2], is more expensive as it requires an extra projection step. Moreover, it could be more difficult to achieve convergence. In order to avoid the aforementioned challenges, we introduce a *re-parameterization* mechanism to remove constraints needed to guarantee stability.

**Theorem 7** (Re-parameterization)**.** *For a separable ARMA layer in Definition 5, if we re-parameterize each tuple* $(f_{-1,t}^{(q)}, f_{1,t}^{(q)}, g_{-1,t}^{(q)}, g_{1,t}^{(q)})$ *as learnable parameters* $(\alpha_{q,t}^f, \beta_{q,t}^f, \alpha_{q,t}^g, \beta_{q,t}^g)$:

$$\begin{pmatrix} f_{-1,t}^{(q)} & g_{-1,t}^{(q)} \\ f_{1,t}^{(q)} & g_{1,t}^{(q)} \end{pmatrix} = \begin{pmatrix} f_{0,t}^{(q)} & 0 \\ 0 & g_{0,t}^{(q)} \end{pmatrix} \begin{pmatrix} \sqrt{2}/2 & -\sqrt{2}/2 \\ \sqrt{2}/2 & \sqrt{2}/2 \end{pmatrix} \begin{pmatrix} \alpha_{q,t}^f & \alpha_{q,t}^g \\ \tanh(\beta_{q,t}^f) & \tanh(\beta_{q,t}^g) \end{pmatrix} \tag{9}$$

*then the layer is stable for* ***arbitrary*** $\{(f_{0,t}^{(q)}, \alpha_{q,t}^f, \beta_{q,t}^f, g_{0,t}^{(q)}, \alpha_{q,t}^g, \beta_{q,t}^g)_{q=1}^Q\}_{t=1}^T$ *with no constraints.*

In practice, we set $f_{0,t}^{(q)} = g_{0,t}^{(q)} = 1$ (since the scale can be learned by the moving-average kernel), and only store and optimize over each tuple $(\alpha_{q,t}^f, \beta_{q,t}^f, \alpha_{q,t}^g, \beta_{q,t}^g)$. In other words, each autoregressive filter $\mathcal{A}_{:,:,t}$ is constructed from $(\alpha_{q,t}^f, \beta_{q,t}^f, \alpha_{q,t}^g, \beta_{q,t}^g)_{q=1}^Q$ on the fly during training or inference.

**Experimental Demonstration of Re-parameterization.** To verify that the re-parameterization mechanism is essential for stable training, we train a VGG-11 network [31] on the CIFAR-10 dataset, where all convolutional layers are replaced by ARMA layers with autoregressive coefficients initialized as zeros. We compare the learning curves using the re-parameterization v.s. not using the re-parameterization in Figure 5. As we can see, the training quickly converges under our proposed re-parameterization mechanism with which the stability of the network is guaranteed. However, without the re-parameterization mechanism, a naive training of the ARMA network never converges and gets NaN error quickly. The experiment thus verifies that the theory in Theorem 7 is effective in guaranteeing stability.

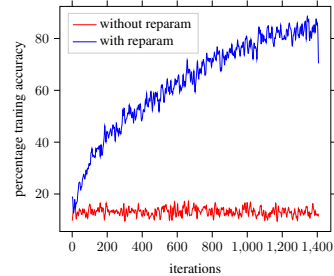

**Figure 5:** Learning curves with and without re-parameterization on an ARMA network with a VGG-11 backbone on CIFAR-10.

## 6 Experiments

We apply our ARMA networks on two dense prediction problems – pixel-level video prediction and semantic segmentation to demonstrate the effectiveness of ARMA networks. **(1)** We incorporate our ARMA layers in U-Nets [30, 36] for semantic segmentation, and in the ConvLSTM network [6, 39] for video prediction. **We show that the resulted ARMA U-Net and ARMA-LSTM models uniformly outperform the baselines on both tasks. (2)** We then interpret the varying performance of ARMA networks on different tasks by visualizing the histograms of the learned autoregressive coefficients. We include the detailed setups (datasets, model architectures, training strategies, and evaluation metrics) and visualization in Appendix A for reproducibility purposes.

**Semantic Segmentation on Biomedical Medical Images.** We evaluate our ARMA U-Net on the lesion segmentation task in ISIC 2018 challenge [38], comparing against a baseline U-Net [30] and non-local U-Net [36] (U-Net augmented with non-local attention blocks).

**Table 2:** **Semantic segmentation on ISIC dataset**. For all metrics (ACC, SE, SP, PC, F1 and JS), higher values indicates better performance. The reported numbers are an average of 10 runs with different seeds.

| Model | params. | ACC | SE | SP | PC | F1 | JS |
|---|---|---|---|---|---|---|---|
| U-Net [30] | 3.453M | 0.946 ± 0.003 | 0.884 ± 0.019 | **0.977** ± 0.005 | 0.857 ± 0.020 | 0.842 ± 0.009 | 0.754 ± 0.011 |
| NL U-Net [36] | 4.403M | 0.945 ± 0.003 | 0.877 ± 0.017 | 0.973 ± 0.004 | 0.844 ± 0.014 | 0.831 ± 0.012 | 0.741 ± 0.013 |
| ARMA U-Net | 3.455M | 0.955 ± 0.003 | 0.896 ± 0.011 | 0.972 ± 0.005 | **0.873** ± 0.011 | 0.861 ± 0.007 | 0.780 ± 0.009 |
| NL ARMA U-Net | 4.405M | **0.960** ± 0.002 | **0.909** ± 0.009 | 0.968 ± 0.004 | 0.870 ± 0.011 | **0.870** ± 0.006 | **0.790** ± 0.008 |

*ARMA networks outperform both baselines in almost all metrics.* As shown in Table 2, our (non-local) ARMA U-Net outperforms both U-Net and non-local U-Net except for specificity (SP). Furthermore, we find that the synergy of non-local attention and ARMA layers achieves the best results among all.

**Pixel-level Video Prediction.** We evaluate our ARMA-LSTM network on the Moving-MNIST-2 dataset [12] with different moving velocities, comparing against the baseline ConvLSTM network [6, 39] and its augmentation using dilated convolutions and non-local attention blocks [34]. As shown in the visualizations in Appendix A, the dilated ARMA-LSTM does not have gridding artifacts as in dilated Conv-LSTM; that is, *ARMA removes the gridding artifacts.*

**Table 3:** 10-frames **video prediction** on Moving-MNIST-2 with three different speeds (results averaged over 10 predicted frames). MA and AR denote the size of moving-average and autoregressive kernels respectively, and dil. denotes dilation in the moving-average kernel. Higher PSNR, SSIM values indicate better performance.

| Model | MA | AR | dil. | params. | original speed PSNR | SSIM | 2X speed PSNR | SSIM | 3X speed PSNR | SSIM |
|---|---|---|---|---|---|---|---|---|---|---|
| Conv-LSTM (size 3) | 3 | 1 | 1 | 0.887M | 18.24 | 0.867 | 16.62 | 0.827 | 15.81 | 0.810 |
| Conv-LSTM (size 5) | 5 | 1 | 1 | 2.462M | 19.58 | 0.901 | 17.61 | 0.856 | 16.99 | 0.841 |
| Dilated Conv-LSTM | 3 | 2 | 2 | 0.887M | 19.16 | 0.893 | 17.92 | 0.858 | 17.48 | 0.846 |
| Dilated ARMA-LSTM | 3 | 3 | 2 | 0.893M | **19.72** | **0.904** | 18.05 | 0.870 | 17.65 | 0.855 |
| ARMA-LSTM (size 3) | 3 | 2 | 1 | 0.893M | **19.72** | 0.899 | **18.73** | **0.881** | **18.13** | **0.869** |

*ARMA networks outperform larger networks:* As shown in Table 3, our ARMA networks with kernel sizes $3 \times 3$ outperform all baselines under all velocities (at the original speed, our ARMA network requires dilated convolutions to achieve the best performance). Moreover, for videos with a higher moving speed, the advantage is more pronounced as expected due to ARMA's ability to expand the ERF. The ARMA networks improve the best baseline (Conv-LSTM with kernel size $5 \times 5$) in PSNR by 6.36% at 2X speed and by 6.70% at 3X speed, with 63.7% fewer parameters.

**Table 4:** **Comparison with non-local attention blocks** on **video prediction**. The original networks are the same as in Table 3. Each non-local network additionally inserts two non-local blocks in the corresponding base network.

| Model | Original PSNR | SSIM | Non-local PSNR | SSIM |
|---|---|---|---|---|
| ConvLSTM (size 3) | 18.24 | 0.867 | 19.45 | 0.895 |
| ConvLSTM (size 5) | 19.58 | **0.901** | 19.18 | 0.891 |
| ARMA-LSTM (size 3) | **19.72** | 0.899 | **19.62** | **0.897** |

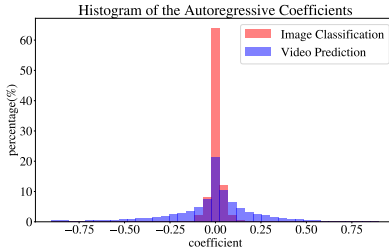

**Figure 6:** Histogram of the autoregressive coefficients in trained ARMA networks.

*ARMA networks outperform non-local blocks:* As shown in Table 4, our ARMA-LSTM with kernel sizes $3 \times 3$ outperforms the Conv-LSTMs augmented by non-local blocks. However, the non-local mechanism does not always improve the baselines or our models. When both ARMA-LSTM and Conv-LSTM are combined with non-local blocks, our model achieves better performance compared to the non-ARMA baselines.

**Interpretation by Autoregressive Coefficients.** Figure 6 compares the histograms of the trained autoregressive coefficients between video prediction and image classification to explain why ARMA networks achieve impressive performance in dense prediction, (subsection A.4 demonstrates the ARMA networks' performance in image classifications with baselines VGG and ResNet.)

1. The histograms demonstrate how ARMA networks adaptively learn autoregressive coefficients according to the tasks. As motivated in the introduction, dense prediction such as video prediction requires each layer to have a large receptive field to capture global information.
2. The large autoregressive coefficients in the video prediction model suggest that the overall ERF is significantly expanded. In the image classification model, global information is already aggregated by pooling (downsampling) layers and a fully-connected classification layer. Therefore, the ARMA layers automatically learn nearly zero autoregressive coefficients.

## 7 Discussion

This paper proposes a novel *ARMA* layer capable of expanding a network's effective receptive field adaptively. Our method is related to techniques in signal processing and machine learning. First, an ARMA layer is equivalent to a multi-channel *impulse response filter* in signal processing [29]. Alternatively, we can interpret the autoregressive layer as a learnable *spectral normalization* [26] following the moving-average layer. Additionally, the ARMA layer is a *linear recurrent neural network*, where the recurrent propagations are over the spatial domain (section 2).

## Funding Disclosure

This research is supported by startup fund from Department of Computer Science of University of Maryland, National Science Foundation IIS-1850220 CRII Award 030742- 00001, DOD-DARPA-Defense Advanced Research Projects Agency Guaranteeing AI Robustness against Deception (GARD), and Laboratory for Physical Sciences at University of Maryland. Huang is also supported by Adobe, Capital One and JP Morgan faculty fellowships.

## Impact Statement

Our presented ARMA layer is a plug-and-play module that can replace any convolutional layer in neural networks. The module is particularly effective in dense prediction problems, including video prediction, object detection and medical image segmentation.

These improved performance in all these applications could revolutionize people's daily life; it could alarm humans potential risks, relieve workers from repeat laboring, and help experts in making better decisions. For examples, video prediction in autonomous system helps to anticipate future risks and contributes to safe self-driving, and medical image segmentation could provide additional information to doctors and help them to make more reliable decisions on high-stakes tasks.

However, these applications also raise controversies in the society. For examples, a faulty prediction in self-driving car or medical diagnostic system could lead to deadly consequence. Furthermore, object detection could be misused for military purposes. To understand and thus mitigate these potential risks, we suggest researchers in engineering and social sciences to investigate questions such as:

- How to systematically verify the capacity of a machine learning model, such that certain behavior can be prohibited before deployment?
- How to define the responsibility if a machine learning system produces an undesired outcome (e.g. car crash or misdiagnosis)?

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
