[Supplementary Material]

# Appendix of ARMA Nets:
# Expanding Receptive Field for Dense Prediction

## A    Supplementary Materials for Experiments

In this section, we explain detailed setups (datasets, model architectures, learning strategies, and evaluation metrics) of all experiments, and provide additional visualizations of the results.

### A.1    Visualization of Effective Receptive Field

In the simulations in subsection 3.2, all linear networks have 32 channels and $64 \times 64$ feature size at each layer. The filter size for both moving-average coefficients and autoregressive coefficients is set to $3 \times 3$: each moving-average kernel $\mathcal{W}$ is initialized using Xavier's method, while the autoregressive kernel $\mathcal{A}$ is initialized randomly within a stable region $-a \leq f_{-1,t}^{(q)} + f_{1,t}^{(q)} \leq 0, -a \leq g_{-1,t}^{(q)} + g_{1,t}^{(q)} \leq 0, \forall t \in [T], q \in [Q]$ (see section 5 for details). We compute each heat map in Figure 3 as an average of 32 gradient maps from different channels.

### A.2    Multi-frame Video Prediction

**Datasets and Metrics**    We generate the Moving-MNIST-2 dataset by moving two digits of size $28 \times 28$ in MNIST dataset within a $64 \times 64$ black canvas [12]. Each digit starts from a random initial location, moves with constant velocity in the canvas, and bounces when they reach the boundary. In addition to the default velocity in the public generator [12], we increase the velocity to $2\times$ and $3\times$ to test all models on videos with stronger motions. For each velocity, we generate 10,000 videos for the training set, 3,000 for the validation set, and 5,000 for the test set, where each video contains 20 frames. We train all models to predict the next 10 frames given 10 input frames, and we evaluate their performance based on the metrics of *mean square error* (MSE), *peak signal-noise ratio* (PSNR) and *structure similarity* (SSIM) [37].

**Model Architectures**    **(1) Baselines.** The backbone architecture consists of a stack of 12 Conv-LSTM modules, and each module contains 32 units (channels). Following [6], two skip connections that perform channel concatenation are added between (3, 9) and (6, 12) module. An additional traditional convolutional layer is applied on top of all recurrent layers to compute the predicted frames. The backbone architecture is illustrated in Figure 7. In the baseline networks, we consider three different convolutions at each layer: **(a)** Traditional convolution with filter size $3 \times 3$; **(b)** Traditional convolution with filter size $5 \times 5$; and **(c)** 2-dilated convolution with filter size $3 \times 3$.

**(2) ARMA networks.** Our ARMA networks use the same backbone architecture as baselines and replace their convolutional layers with ARMA layers. For all ARMA models, we set the filter size for both moving-average and autoregressive parts to $3 \times 3$. In the ARMA networks, we consider two different convolutions each layer: **(a)** The moving-average part is a traditional convolution; **(b)** We further consider using 2-dilated convolution in the moving-average part.

**(3) Non-local networks.** In non-local networks, we additionally insert two non-local blocks in the backbone architecture, as illustrated in Figure 8. In each non-local block, we use embedded Gaussian as the non-local operation [34], and we replace the batch normalization by instance normalization that is compatible with recurrent neural networks. In non-local networks, we consider three types of convolutions at each layer: **(1)(2)** Traditional convolutions with filter size $3 \times 3$ and $5 \times 5$; **(3)** ARMA layer with $3 \times 3$ moving-average and autoregressive filters.

**Figure 7:** Conv(ARMA)-LSTM.          **Figure 8:** Non-Local Conv(ARMA)-LSTM.

**Training Strategy**    All models are trained using ADAM optimizer [18], with $\mathcal{L}_1 + \mathcal{L}_2$ loss and for 500 epochs. We set the initial learning rate to $10^{-3}$, and the value for gradient clipping to 3. Learning

rate decay and scheduled sampling [1] are used to ease training. Scheduled sampling starts once the model does not improve in 20 epochs (in term of validation loss), and the sampling ratio is decreased linearly by $4 \times 10^{-4}$ each epoch (i.e., scheduling sampling lasts for 250 epochs). Learning rate decay is further activated if the validation loss does not drop in 20 epochs, and the learning rate is decreased exponentially by 0.98 every 5 epochs. All convolutional layers and moving-average parts in ARMA layers are initialized by Xavier's normalized initializer [13], and autoregressive coefficients in ARMA layers are initialized as zeros (i.e., each ARMA layer is initialized as a traditional layer).

**Visualization of the Predictions** We visualize the predictions by different models under three moving velocites in Figure 9, Figure 10 and Figure 11 Notice that the gridding artifacts by dilated convolutions are removed by ARMA layer: since each neuron receives information from all pixels in a local region (Figure 2d), adjacent neurons are on longer computed from separate sets of pixels. Moreover, for videos with a higher moving speed, the advantage of our ARMA layer is more pronounced as expected due to ARMA's ability to expand the ERF.

Figure 9: **Prediction on Moving-MNIST-2 (original speed)**. The first row contains the last 3 input frames and 10 ground-truth frames for models to predict.

Figure 10: **Prediction on Moving-MNIST-2 ($2\times$ speed)**. The first row contains the last 3 input frames and 10 ground-truth frames for models to predict.

**Figure 11: Prediction on Moving-MNIST-2 (3× speed).** The first row contains the last 3 input frames and 10 ground-truth frames for models to predict.

**(a)** MSE      **(b)** PSNR      **(c)** SSIM

**Figure 12: Per-frame performance comparison** of our ARMA and our dilated ARMA networks v.s. the Conv-LSTM, dilated Conv-LSTM baselines for Moving-MNIST-2 (original speed). Lower MSE values (in $10^{-3}$) or higher PSNR/SSIM values indicate better performance.

**(a)** MSE      **(b)** PSNR      **(c)** SSIM

**Figure 13: Per-frame performance comparison** of our ARMA and our dilated ARMA networks v.s. the Conv-LSTM, dilated Conv-LSTM baselines for Moving-MNIST-2 (2× speed). Lower MSE values (in $10^{-3}$) or higher PSNR/SSIM values indicate better performance.

**(a)** MSE          **(b)** PSNR          **(c)** SSIM

**Figure 14: Per-frame performance comparison** of our ARMA and our dilated ARMA networks v.s. the Conv-LSTM, dilated Conv-LSTM baselines for Moving-MNIST-2 ($3\times$ speed). Lower MSE values (in $10^{-3}$) or higher PSNR/SSIM values indicate better performance.

## A.3 Medical Image Segmentation

To demonstrate ARMA networks' applicability to image segmentation, we evaluate it on a challenging medical image segmentation problem.

**Dataset and Metrics** For all experiments, we use a dataset from ISIC 2018: Skin Lesion Analysis Towards Melanoma Detection [9], which can be downloaded online[1]. In this task, a model aims to predict a binary mask that indicates the location of the primary skin lesion for each input image. The dataset consists of 2594 images, and we resize each image to $224 \times 224$. We split the dataset into a training set, validation set and test set with ratios of $0.7$, $0.1$, and $0.2$, respectively. All models are evaluated using the following metrics: $AC = (TP + TN)/(TP + TN + FP + FN)$, $SE = TP/(TP+FN)$, $SP = TN/(TN+FP)$, $PC = TP/(TP+FP)$, $F1 = 2PC \cdot SE/(PC+SE)$ and $JS = |GT \cap SR|/|GT \cup SR|$, where TP stands for true positive, TN for true negative, FP for false positive, FN for false negative, GT for ground truth mask and SR for predictive mask.

**Model Architectures** **(1) Baselines.** We use U-Net [30] and non-local U-Net [36] as baseline models. U-Net has a contracting path to capture context, and a symmetric expanding path enables precise localization. The network architecture is illustrated in Figure 15a. Non-local U-Net further contains dglobal aggregation blocks based on the self-attention operator to aggregate global information without a deep encoder for biomedical image segmentation, as illustrated in Figure 15b. **(2) Our architectures.** We replace all traditional convolution layers with ARMA layers in U-Net and non-local U-Net.

**(a)** U-Net architecture.          **(b)** Non-local U-Net architecture.

**Figure 15:** Backbone architectures.

**Training Strategy.** All models are trained using ADAM optimizer [18] with binary cross entropy (BCE) loss. For initial learning rate, we search from $10^{-2}$ to $10^{-5}$ and choose $10^{-3}$ for U-Net and

$10^{-2}$ for non-local U-Net. The learning rate is decayed by $0.98$ every epoch. During training, each image is randomly augmented by rotation, cropping, shifting, color jitter, and normalization following the public source code[2].

| Input image | Ground truth | U-net **with ARMA** | U-net without ARMA | Non-Local U-net **with ARMA** | Non-Local U-net without ARMA |

**Figure 16:** Predictive results of U-Net and Non-local U-Net with/without ARMA layers.

## A.4 Image Classification

**Model Architectures and Datasets** We replace the traditional convolutional layers by ARMA layers in three benchmarking architectures for image classification: AlexNet [19], VGG-11 [31], and ResNet-18 [14]. We apply our proposed ARMA networks on CIFAR10 and CIFAR100 datasets. Both datasets have $50000$ training examples and $10000$ test examples, and we use $5000$ examples from the training set for validation (and leave $45000$ examples for training).

**Training Strategy** We train all models using cross-entropy loss and SGD optimizer with batch size $128$, learning rate $0.1$, weight decay $0.0005$ and momentum $0.9$. For CIFAR10, the models are trained for $300$ epochs and we half the learning rate every $30$ epochs. For CIFAR100, the models are trained for $200$ epochs and we divide the learning rate by $5$ at the $60^{th}$, $120^{th}$, $160^{th}$ epochs.

**Results** The experimental results are summarized in Table 5. Our results show that ARMA models achieve comparable or slightly better results than the benchmarking architectures. Replacing the traditional convolutional layer with our proposed ARMA layer slightly boosts VGG-11 and ResNet-18 by 0.01%-0.1% in terms of accuracy. **Since image classifications tasks do not require convolutional layers to have large receptive fields, the learned autoregressive coefficients concentrate around 0, as shown in Figure 6.** Consequently, ARMA networks effectively reduce to traditional convolutional neural networks and therefore achieve comparable results.

| | AlexNet | | VGG-11 | | ResNet-18 | |
|---|---|---|---|---|---|---|
| | Conv. | ARMA | Conv. | ARMA | Conv. | ARMA |
| CIFAR10 | $86.30 \pm 0.29$ | $85.67 \pm 0.19$ | $91.57 \pm 0.59$ | $91.57 \pm 0.73$ | $95.01 \pm 0.15$ | $95.07 \pm 0.13$ |
| CIFAR100 | $58.99 \pm 0.37$ | $57.43 \pm 0.24$ | $68.25 \pm 0.11$ | $68.36 \pm 1.67$ | $73.71 \pm 0.23$ | $73.72 \pm 0.52$ |

**Table 5: Image classification** on CIFAR10 and CIFAR100. We report accuracy (%) and standard deviations from 10 runs with different seeds. Since image classifications tasks do not require convolutional layers to have large receptive fields, the learned autoregressive coefficients are highly concentrated around 0, as shown in Figure 6. Consequently, ARMA networks effectively reduce to traditional CNNs and therefore achieve comparable results.

# B  Analysis of Effective Receptive Field (ERF)

In this section, we prove Theorem 3 in subsection 3.2. Here, we use $\boldsymbol{a}^{(\ell)} = \{\cdots, a_{-1}^{(\ell)}, a_0^{(\ell)}, a_1^{(\ell)}, \cdots\}$ to denote the $\ell^{\text{th}}$ layer's autoregressive (AR) coefficients, and $\boldsymbol{w}^{(\ell)} = \{\cdots, w_{-1}^{(\ell)}, w_0^{(\ell)}, w_1^{(\ell)}, \cdots\}$ the $\ell^{\text{th}}$ layer's moving-average (MA) coefficients.

## B.1  ERF of General Linear Convolutional Networks

The proof of Theorem 3 is based on the following theorem on linear convolutional networks [25], which includes both CNN and ARMA networks as special cases.

**Theorem 8** (**ERF of Linear Convolutional Networks with Infinite Horizon**). *Consider an L-layer linear convolutional network (without activation and pooling layers), where its $\ell^{th}$-layer computes a weighted-sum of its input $y_i^{(\ell)} = \sum_{p=-\infty}^{+\infty} w_p^{(\ell)} y_{i-p}^{(\ell-1)}$. Suppose the weights are non-negative $w_p^{(\ell)} \geq 0$ and normalized at each layer $\sum_{p=-\infty}^{+\infty} w_p^{(\ell)} = 1$, the network has an ERF radius as*

$$r(ERF)^2 = \sum_{\ell=1}^{L} \left[ \sum_{p=-\infty}^{+\infty} p^2 w_p^{(\ell)} - \left( \sum_{p=-\infty}^{+\infty} p w_p^{(\ell)} \right)^2 \right]. \tag{B.1}$$

*Furthermore, the ERF converges to a Gaussian density function when L tends to infinity.*

*Proof.* In this linear convolutional network, the gradient maps can be computed with chain rule as

$$g_{i,:} = \boldsymbol{w}^{(1)^\top} * \boldsymbol{w}^{(2)^\top} \cdots * \boldsymbol{w}^{(L)^\top}, \ \forall i \in \mathbb{Z}, \tag{B.2}$$

where $\boldsymbol{w}^{(\ell)^\top}$ denotes the reversed version of $\boldsymbol{w}^{(\ell)}$. Notice that **(1)** The gradient maps do not depend on the input, i.e., they are data-independent; **(2)** The gradient maps are identical across different locations in the output. Consequently, the ERF is equal to any one gradient map above

$$\text{ERF} = \boldsymbol{w}^{(1)^\top} * \boldsymbol{w}^{(2)^\top} \cdots * \boldsymbol{w}^{(L)^\top}. \tag{B.3}$$

The remaining part of the proof makes use of the *probabilistic method*, which interprets the operation at each layer as a discrete random variable. Since the weights at each layer are non-negative and normalized, we interpret them as values of a probability mass function. Concretely, we construct $L$ independent random variables $\{W^{(1)}, \cdots, W^{(L)}\}$ such that $\mathbb{P}[W^{(\ell)} = p] = w_{-p}^{(\ell)}$. Similarly, we introduce a random variable $S^{(L)}$ to represent the ERF, i.e., $\mathbb{P}[S^{(L)} = p] = \text{ERF}_p$. As a result, the ERF radius is equal to standard deviation of $S^{(L)}$, or equivalently $r(\text{ERF})^2 = \mathbb{V}[S^{(L)}]$.

Recall that *addition of independent random variables results in convolution of their probability mass functions*, Equation B.3 implies that $S^{(L)}$ is an addition of all $W^{(\ell)}$'s, i.e., $S^{(L)} = \sum_{\ell=1}^{L} W^{(\ell)}$. Therefore, the variance of $S^{(L)}$ is equal to a summation of the variances for $W^{(\ell)}$'s. Thus,

$$\mathbb{V}[S^{(L)}] = \sum_{\ell=1}^{L} \mathbb{V}[W^{(\ell)}] = \sum_{\ell=1}^{L} \left[ \mathbb{E}[(W^{(\ell)})^2] - \mathbb{E}[W^{(\ell)}]^2 \right] \tag{B.4}$$

$$= \sum_{\ell=1}^{L} \left[ \sum_{p=-\infty}^{+\infty} p^2 w_p^{(\ell)} - \left( \sum_{p=-\infty}^{+\infty} p w_p^{(\ell)} \right)^2 \right], \tag{B.5}$$

which proves the Equation B.1. Furthermore, the *Lyapunov central limit theorem* shows that $(S^{(L)} - \mathbb{E}[S^{(L)}])/\mathbb{V}[S^{(L)}]$ converges to a standard normal random variable if $L$ tends to infinity

$$\frac{S^{(L)} - \mathbb{E}[S^{(L)}]}{\sqrt{\mathbb{V}[S^{(L)}]}} = \frac{\sum_{\ell=1}^{L} \left( W^{(\ell)} - \mathbb{E}[W^{(\ell)}] \right)}{\sqrt{\sum_{\ell=1}^{L} \mathbb{V}[W^{(\ell)}]}} \xrightarrow{D} \mathcal{N}(0, 1). \tag{B.6}$$

That is, the ERF function is approximately Gaussian when the number of layers $L$ is large enough. $\quad\square$

**B.2   ERF of Traditional CNNs ($a_1^{(\ell)} = -a^{(\ell)} = 0$)**

As a warmup, we first provide a proof for the special case of traditional CNN where $a^{(\ell)} = 0$ for all layers. For reference, we list the first two cases of *Faulhaber's formula*:

$$\sum_{p=0}^{K-1} p = \frac{K(K-1)}{2}, \tag{B.7a}$$

$$\sum_{p=0}^{K-1} p^2 = \frac{K(K-1)(2K-1)}{6}. \tag{B.7b}$$

*Proof.* In this special case, we can plug $w_p^{(\ell)} = 1/K^{(\ell)}$ for $p = 0, d^{(\ell)}, \cdots, d^{(\ell)}(K^{(\ell)} - 1)$ into Equation B.1 directly to obtain ERF radius.

$$r(\text{ERF})^2 = \sum_{\ell=1}^{L} \left[ \sum_{p=0}^{K^{(\ell)}-1} \frac{\left(pd^{(\ell)}\right)^2}{K^{(\ell)}} - \left( \sum_{p=0}^{K^{(\ell)}-1} \frac{pd^{(\ell)}}{K^{(\ell)}} \right)^2 \right] \tag{B.8}$$

$$= \sum_{\ell=1}^{L} \left[ \frac{d^{(\ell)^2}}{K^{(\ell)}} \frac{K^{(\ell)} \left(K^{(\ell)} - 1\right) \left(2K^{(\ell)} - 1\right)}{6} - \left( \frac{d^{(\ell)}}{K^{(\ell)}} \frac{K^{(\ell)} \left(K^{(\ell)} - 1\right)}{2} \right)^2 \right] \tag{B.9}$$

$$= \sum_{\ell=1}^{L} \frac{d^{(\ell)^2} \left(K^{(\ell)^2} - 1\right)}{12}, \tag{B.10}$$

where the infinite series are computed using Equation B.7a and Equation B.7b. Taking square root on both sides completes the proof for the special case of CNNs. □

**ERF Analysis of CNNs.** If we further assume that all layers are identical, i.e., $K^{(\ell)} = K, d^{(\ell)} = d$ for $1 \leq \ell \leq L$, we can simplify Equation B.10 as

$$r(\text{ERF}) = \sqrt{L} \cdot \sqrt{\frac{d^2(K^2-1)}{12}} = O\left(dK\sqrt{L}\right). \tag{B.11}$$

That is, the ERF radius grows linearly with the kernel size $K$ and the dilation $d$, but sub-linearly with the number of layers $L$ in the linear network.

**B.3   ERF of ARMA networks ($a_1^{(\ell)} = -a^{(\ell)} \leq 0$)**

In the part, we provide a proof for general ARMA networks where $a^{(\ell)} \leq 0$. In sketch, the proof consists of three steps: **(1)** we introduce *inverse convolution* and convert each ARMA model to a moving-average model: $\boldsymbol{a} * \boldsymbol{y} = \boldsymbol{w} * \boldsymbol{x} \implies \boldsymbol{y} = \boldsymbol{f} * \boldsymbol{x}$, where $\boldsymbol{f}$ represents a convolution with *infinite* number of coefficients, $\boldsymbol{x}$ and $\boldsymbol{y}$ are the input and output of the model respectively. **(2)** We derive the *moment generating function* (MGF) of the moving-average coefficients from the first step, and use the functions to compute the first and second moments. **(3)** We plug the moments from the second step into Equation B.1 to obtain Equation 4.

**Definition 9** (**Inverse Convolution**). *Given a convolution (with coefficients) $\boldsymbol{a}$, its inverse convolution $\overline{\boldsymbol{a}}$ is defined such that $\boldsymbol{a} * \overline{\boldsymbol{a}} = \overline{\boldsymbol{a}} * \boldsymbol{a} = \boldsymbol{\delta}$ is an identical mapping, i.e.,*

$$\sum_{p=-\infty}^{+\infty} a_{i-p}\overline{a}_p = \delta_i = \begin{cases} 1 & i = 0 \\ 0 & i \neq 0 \end{cases} \tag{B.12}$$

*Remark:* The inverse convolution does not exist for any convolution $\boldsymbol{a}$. A necessary and sufficient condition for invertibility of $\boldsymbol{a}$ is that its Fourier transform is non-zero everywhere [29].

**Definition 10** (**Moments and Moment Generating Function, MGF**). *Given a convolution (with coefficients)* $\boldsymbol{f}$, *its* $i^{th}$ *moment is defined as*

$$M_i(\boldsymbol{f}) = \sum_{p=-\infty}^{+\infty} f_p p^i. \tag{B.13}$$

*Furthermore, we define the moment generating function of the coefficients* $\boldsymbol{f}$ *as*

$$M_{\boldsymbol{f}}(\lambda) = \sum_{p=-\infty}^{+\infty} f_p e^{\lambda p}. \tag{B.14}$$

*The name "moment generating" comes from the fact that*

$$M_i(\boldsymbol{f}) = \left. \frac{d^i M_{\boldsymbol{f}}(\lambda)}{d\lambda^i} \right|_{\lambda=0}. \tag{B.15}$$

*Remark:* Since moment generating function (MGF) could be interpreted as a real-valued discrete-time Fourier transform (DTFT), the properties of MGF are very similar to the ones of DTFT. In particular, the convolution theorem also holds for MGF, i.e., $M_{\boldsymbol{f}*\boldsymbol{g}}(\lambda) = M_{\boldsymbol{f}}(\lambda)M_{\boldsymbol{g}}(\lambda)$. If two convolutions $\boldsymbol{a}$ and $\overline{\boldsymbol{a}}$ are inverse to each other, we have $M_{\boldsymbol{a}}(\lambda)M_{\overline{\boldsymbol{a}}}(\lambda) = 1$.

Now we are ready to prove Theorem 3 using Theorem 8 and Definitions 9 and 10.

*Proof.* Let $\boldsymbol{f}^{(\ell)} = \overline{\boldsymbol{a}}^{(\ell)} * \boldsymbol{w}^{(\ell)}$, we have

$$\boldsymbol{y}^{(\ell)} = \boldsymbol{\delta} * \boldsymbol{y}^{(\ell)} = \left( \overline{\boldsymbol{a}}^{(\ell)} * \boldsymbol{a}^{(\ell)} \right) * \boldsymbol{y}^{(\ell)} = \overline{\boldsymbol{a}}^{(\ell)} * \left( \boldsymbol{a}^{(\ell)} * \boldsymbol{y}^{(\ell)} \right)$$
$$= \overline{\boldsymbol{a}}^{(\ell)} * \left( \boldsymbol{w}^{(\ell)} * \boldsymbol{y}^{(\ell-1)} \right) = \left( \overline{\boldsymbol{a}}^{(\ell)} * \boldsymbol{w}^{(\ell)} \right) * \boldsymbol{y}^{(\ell-1)} = \boldsymbol{f}^{(\ell)} * \boldsymbol{y}^{(\ell-1)}, \tag{B.16}$$

where each $\boldsymbol{f}^{(\ell)}$ has infinite number of coefficients. We denote the MGF of $\boldsymbol{f}^{(\ell)}$ as $M_{\boldsymbol{f}^{(\ell)}}$, and its first and second moments as $M_1(\boldsymbol{f}^{(\ell)})$ and $M_2(\boldsymbol{f}^{(\ell)})$. With the moments of $\boldsymbol{f}^{(\ell)}$, we can rewrite Equation B.1 in Theorem 8 as

$$r(\text{ERF})^2 = \sum_{\ell=1}^{L} \left[ M_2(\boldsymbol{f}^{(\ell)}) - \left( M_1(\boldsymbol{f}^{(\ell)}) \right)^2 \right]. \tag{B.17}$$

The remaining part is to compute $M_{\boldsymbol{f}^{(\ell)}}$ for each $\boldsymbol{f}^{(\ell)}$, with which $M_1(\boldsymbol{f}^{(\ell)})$ and $M_2(\boldsymbol{f}^{(\ell)})$ are generated. Notice that $\boldsymbol{f}^{(\ell)} = \overline{\boldsymbol{a}}^{(\ell)} * \boldsymbol{w}^{(\ell)}$ is a convolution between $\overline{\boldsymbol{a}}^{(\ell)}$ and $\boldsymbol{w}^{(\ell)}$, we have

$$M_{\boldsymbol{f}^{(\ell)}}(\lambda) = M_{\overline{\boldsymbol{a}}^{(\ell)}}(\lambda)M_{\boldsymbol{w}^{(\ell)}}(\lambda) = \frac{M_{\boldsymbol{w}^{(\ell)}}(\lambda)}{M_{\boldsymbol{a}^{(\ell)}}(\lambda)} \tag{B.18}$$

$$= \frac{1}{1 - a^{(\ell)}e^\lambda} \sum_{p=0}^{K^{(\ell)}-1} \frac{1 - a^{(\ell)}}{K^{(\ell)}} e^{\lambda p d^{(\ell)}}, \tag{B.19}$$

where the first equation uses the property that $M_{\boldsymbol{a}^{(\ell)}}(\lambda)M_{\overline{\boldsymbol{a}}^{(\ell)}}(\lambda) = 1$ for any $\lambda$. The first moment $M_1(\boldsymbol{f}^{(\ell)})$ is therefore

$$M_1(\boldsymbol{f}^{(\ell)}) = \left. \frac{dM_{\boldsymbol{f}^{(\ell)}}(\lambda)}{d\lambda} \right|_{\lambda=0} \tag{B.20}$$

$$= \left. \frac{a^{(\ell)}}{\left(1 - a^{(\ell)}\lambda\right)^2} \sum_{p=0}^{K^{(\ell)}-1} \frac{1 - a^{(\ell)}}{K^{(\ell)}} e^{\lambda p d^{(\ell)}} + \frac{1}{1 - a^{(\ell)}e^\lambda} \sum_{p=0}^{K^{(\ell)}-1} \frac{1 - a^{(\ell)}}{K^{(\ell)}} p d^{(\ell)} e^{\lambda p d^{(\ell)}} \right|_{\lambda=0} \tag{B.21}$$

$$= \frac{a^{(\ell)}}{1 - a^{(\ell)}} + \frac{d^{(\ell)}}{K^{(\ell)}} \left( \sum_{p=0}^{K^{(\ell)}-1} p \right) \tag{B.22}$$

$$= \frac{a^{(\ell)}}{1 - a^{(\ell)}} + \frac{d^{(\ell)} \left( K^{(\ell)} - 1 \right)}{2}, \tag{B.23}$$

where the last equation makes use of Equation B.7a. Similarly, the second moment $M_2(\boldsymbol{f}^{(\ell)})$ is

$$M_2(\boldsymbol{f}^{(\ell)}) = \left.\frac{d^2 M_{\boldsymbol{f}^{(\ell)}}(\lambda)}{d\lambda^2}\right|_{\lambda=0} \tag{B.24}$$

$$= \frac{a^{(\ell)^2}}{\left(1-a^{(\ell)}\right)^3} \sum_{p=0}^{K^{(\ell)}-1} \frac{1-a^{(\ell)}}{K^{(\ell)}} e^{\lambda p d^{(\ell)}} + \frac{2a^{(\ell)}}{\left(1-a^{(\ell)}e^\lambda\right)^2} \sum_{p=0}^{K^{(\ell)}-1} \frac{1-a^{(\ell)}}{K^{(\ell)}} p d^{(\ell)} e^{\lambda p d^{(\ell)}}$$

$$\left. + \frac{1}{1-a^{(\ell)}e^\lambda} \sum_{p=0}^{K^{(\ell)}-1} \frac{1-a^{(\ell)}}{K^{(\ell)}} \left(p d^{(\ell)}\right)^2 e^{\lambda p d^{(\ell)}}\right|_{\lambda=0} \tag{B.25}$$

$$= \left(\frac{a^{(\ell)}}{1-a^{(\ell)}}\right)^2 + \frac{2a^{(\ell)}}{1-a^{(\ell)}} \frac{d^{(\ell)}}{K^{(\ell)}} \left(\sum_{p=0}^{K^{(\ell)}-1} p\right) + \frac{d^{(\ell)^2}}{K^{(\ell)}} \left(\sum_{p=0}^{K^{(\ell)}-1} p^2\right) \tag{B.26}$$

$$= \left(\frac{a^{(\ell)}}{1-a^{(\ell)}}\right)^2 + \frac{2a^{(\ell)}}{1-a^{(\ell)}} \frac{d^{(\ell)}\left(K^{(\ell)}-1\right)}{2} + \frac{d^{(\ell)^2}\left(K^{(\ell)}-1\right)\left(2K^{(\ell)}-1\right)}{6}. \tag{B.27}$$

Plugging Equation B.23 and Equation B.27 into Equation B.17, we have

$$r(\text{ERF})^2 = \sum_{\ell=1}^{L} \left[\frac{d^{(\ell)^2}\left(K^{(\ell)}-1\right)^2}{12} + \frac{a^{(\ell)}}{\left(1-a^{(\ell)}\right)^2}\right]. \tag{B.28}$$

Taking square root on both sides completes the proof. $\qquad\square$

**ERF Analysis of ARMA Networks.** If we assume all layers are identical, i.e., $K^{(\ell)} = K, d^{(\ell)} = d, a^{(\ell)} = a$ for $1 \le \ell \le L$, we can simplify Equation B.28 as

$$r(\text{ERF}) = \sqrt{L} \cdot \sqrt{\frac{d^2(K^2-1)}{12} + \frac{a}{(1-a)^2}} = O\left(\sqrt{L}\max\left(dK, \frac{\sqrt{a}}{1-a}\right)\right). \tag{B.29}$$

The ERF radius is dominated by the AR coefficient when $a \lessapprox 1$ regardless of the kernel size $K$ and the dilation $d$. The radius still grows sub-linearly with the number of layers $L$.

## C  Computation of ARMA Layers

In the section, we first derive the backpropagation rules in Theorem 4. We then show how to efficiently compute both forward and backward passes in ARMA layer using *Fast Fourier Transform*.

### C.1  Backpropagation in ARMA Models

In this part, we will prove a general theorem for backpropagation in ARMA models. To keep the notations simple, we derive the backpropagation equations for ARMA models with one dimension input/output and one channel. However, the proof techniques can be trivially extended to general ARMA models with high-dimensional input/output with multiple channels.

**Theorem 11** (**Backpropagation in an ARMA Model**). *Consider an ARMA model $\boldsymbol{a} * \boldsymbol{y} = \boldsymbol{w} * \boldsymbol{x}$, where $\boldsymbol{a}$ and $\boldsymbol{w}$ are the sequences of moving-average and autoregressive coefficients respectively, the gradients $\{\partial\mathcal{L}/\partial\boldsymbol{x}, \partial\mathcal{L}/\partial\boldsymbol{w}, \partial\mathcal{L}/\partial\boldsymbol{a}\}$ can be computed from $\partial\mathcal{L}/\partial\boldsymbol{y}$ with the following equations:*

$$\boldsymbol{a}^\top * \frac{\partial\mathcal{L}}{\partial\boldsymbol{x}} = \boldsymbol{w}^\top * \frac{\partial\mathcal{L}}{\partial\boldsymbol{y}}, \tag{C.1a}$$

$$-\boldsymbol{a}^\top * \frac{\partial\mathcal{L}}{\partial\boldsymbol{a}} = \boldsymbol{y}^\top * \frac{\partial\mathcal{L}}{\partial\boldsymbol{y}}, \tag{C.1b}$$

$$\boldsymbol{a}^\top * \frac{\partial\mathcal{L}}{\partial\boldsymbol{w}} = \boldsymbol{x}^\top * \frac{\partial\mathcal{L}}{\partial\boldsymbol{y}}, \tag{C.1c}$$

*where $\boldsymbol{a}^\top$, $\boldsymbol{w}^\top$, and $\boldsymbol{y}^\top$ denote the reversed sequences of $\boldsymbol{a}$, $\boldsymbol{w}$, and $\boldsymbol{y}$ respectively.*

Notice that Theorem 4 is special case of Theorem 11, where Equation 6a is proved by Equation C.1b and Equation 6b is proved by Equation C.1a.

We provide two different proofs of Theorem 11. **(1)** The analysis in our first proof only uses real numbers and applicable to arbitrary convolution types. **(2)** If the convolution is *circular* (as in the implementation of this paper), we provide a simplified proof using Fourier transform (therefore complex numbers). The second proof also suggests an FFT-based algorithm to compute the backpropagation in Equation 4 efficiently.

### C.1.1 Proof in Real Numbers $\mathbb{R}$

Before we prove the theorem, we first prove a useful lemma on the inverse of transposed convolution.

**Lemma 12** (**Inverse of transposed convolution**). *Given a convolution (with coefficients) $\boldsymbol{a}$, the operations of inversion and transposition are exchangeable,*

$$\overline{\boldsymbol{a}^\top} = \overline{\boldsymbol{a}}^\top. \tag{C.2}$$

*In other words, the inverse transposed convolution is equal to the transposed inverse convolution.*

*Proof.* The lemma is an immediate result of the definitions of inverse and transposed convolutions.

$$\sum_{p=-\infty}^{+\infty} a_p^\top \overline{a}_{i-p}^\top = \sum_{p=-\infty}^{+\infty} a_{-p} \overline{a}_{p-i} = \delta_{-i} = \delta_i \quad \forall i, \tag{C.3}$$

which shows the inverse of $\boldsymbol{a}^\top$, i.e., $\overline{\boldsymbol{a}^\top}$, is equal to $\overline{\boldsymbol{a}}^\top$. $\square$

Now we are ready to prove Theorem 11 at the beginning of this section.

*Proof.* To begin with, we write the ARMA model $\boldsymbol{a} * \boldsymbol{y} = \boldsymbol{w} * \boldsymbol{x}$ in its weighted-sum form:

$$\sum_{q=-\infty}^{+\infty} a_q y_{i-q} = \sum_{p=-\infty}^{+\infty} w_p x_{i-p}, \ \forall i. \tag{C.4}$$

Taking derivative w.r.t. $a_r$ on both sides, and since the right side is a constant w.r.t. $a_r$, we have

$$\frac{\partial \left( \sum_{q=-\infty}^{+\infty} a_q y_q \right)}{\partial a_r} = 0, \ \forall i, r. \tag{C.5}$$

By *implicit function theorem*, the left hand side can be further expanded as

$$\frac{\partial \left( \sum_{q=-\infty}^{+\infty} a_q y_{i-q} \right)}{\partial a_r} = \sum_{q=-\infty}^{+\infty} \frac{\partial (a_q y_{i-q})}{\partial a_r} \tag{C.6}$$

$$= \sum_{q \neq r} a_q \frac{\partial y_{i-q}}{\partial a_r} + \left( y_{i-r} + a_r \frac{\partial y_{i-r}}{\partial a_r} \right) \tag{C.7}$$

$$= \sum_{q=-\infty}^{+\infty} a_q \frac{\partial y_{i-q}}{\partial a_r} + y_{i-r} = 0, \ \forall i, r. \tag{C.8}$$

Rearranging the equation above, we have

$$- \sum_{q=-\infty}^{+\infty} a_q \frac{\partial y_{i-q}}{\partial a_r} = y_{i-r}, \ \forall i, r. \tag{C.9a}$$

Repeating the procedure twice for the derivatives w.r.t. $w_r$ and $x_r$, we have two similar equations:

$$\sum_{q=-\infty}^{+\infty} a_q \frac{\partial y_{i-q}}{\partial w_r} = x_{i-r}, \ \forall i, r, \tag{C.9b}$$

$$\sum_{q=-\infty}^{+\infty} a_q \frac{\partial y_{i-q}}{\partial x_r} = a_{i-r}, \ \forall i, r. \tag{C.9c}$$

Since Equation C.9a, Equation C.9b and Equation C.9c take the same form, we only precede with Equation C.9b and obtain $\partial \mathcal{L} / \partial \boldsymbol{w}$. The other two can be derived using the same arguments.

Notice that Equation C.9b can be rewritten as

$$\sum_{q=-\infty}^{+\infty} a_{i-q} \frac{\partial y_q}{\partial w_r} = x_{i-r}, \ \forall i, r \tag{C.10}$$

by changing variable $q$ to $i - q$. Since Equation C.10 holds for any $i$, we further introduce a new index $l$ and change $i$ to $i - l$ on both hand sides:

$$\sum_{q=-\infty}^{+\infty} a_{i-q-l} \frac{\partial y_q}{\partial w_r} = x_{i-r-l}, \ \forall i, r, l. \tag{C.11}$$

Now we convolve both hand sides with $\overline{\boldsymbol{a}}$, the inverse of $\boldsymbol{a}$. Then for all $i$ and $r$, we have

$$\sum_{l=-\infty}^{+\infty} \overline{a}_l \left( \sum_{q=-\infty}^{+\infty} a_{i-q-l} \frac{\partial y_q}{\partial w_r} \right) = \sum_{l=-\infty}^{+\infty} \overline{a}_l x_{i-r-l} \tag{C.12}$$

$$\sum_{q=-\infty}^{+\infty} \left( \sum_{l=-\infty}^{+\infty} \overline{a}_l a_{i-q-l} \right) \frac{\partial y_q}{\partial w_r} = \sum_{l=-\infty}^{+\infty} \overline{a}_l x_{i-r-l} \tag{C.13}$$

$$\frac{\partial y_i}{\partial w_r} = \sum_{q=-\infty}^{+\infty} \delta_{i-q} \frac{\partial y_q}{\partial w_r} = \sum_{l=-\infty}^{+\infty} \overline{a}_l x_{i-r-l}. \tag{C.14}$$

Subsequently, we apply the chain rule to obtain $\partial \mathcal{L} / \partial w_r$

$$\frac{\partial \mathcal{L}}{\partial w_r} = \sum_{i=-\infty}^{+\infty} \frac{\partial y_i}{\partial w_r} \frac{\partial \mathcal{L}}{\partial y_i} = \sum_{i=-\infty}^{+\infty} \sum_{l=-\infty}^{+\infty} \overline{a}_l x_{i-r-l} \frac{\partial \mathcal{L}}{\partial y_i}, \ \forall r. \tag{C.15}$$

Finally, we convolve both hand sides with $\boldsymbol{a}^\top$, the transpose of $\boldsymbol{a}$, to obtain the ARMA form of backpropagation rule.

$$\sum_{r=-\infty}^{+\infty} a_{j-r}^\top \frac{\partial \mathcal{L}}{\partial w_r} = \sum_{r=-\infty}^{+\infty} a_{j-r}^\top \left( \sum_{i=-\infty}^{+\infty} \sum_{l=-\infty}^{+\infty} \overline{a}_l x_{i-r-l} \frac{\partial \mathcal{L}}{\partial y_i} \right) \tag{C.16}$$

$$= \sum_{r=-\infty}^{+\infty} \sum_{i=-\infty}^{+\infty} \sum_{l=-\infty}^{+\infty} a_{j-r}^\top \overline{a}_l x_{i-r-l} \frac{\partial \mathcal{L}}{\partial y_i} \tag{C.17}$$

$$= \sum_{r=-\infty}^{+\infty} \sum_{i=-\infty}^{+\infty} \sum_{l=-\infty}^{+\infty} a_{j-r}^\top \overline{a}_{l-r} x_{i-l} \frac{\partial \mathcal{L}}{\partial y_i} \tag{C.18}$$

$$= \sum_{r=-\infty}^{+\infty} \sum_{i=-\infty}^{+\infty} \sum_{l=-\infty}^{+\infty} a_{j-r}^\top \overline{a}_{r-l}^\top x_{l-i}^\top \frac{\partial \mathcal{L}}{\partial y_i} \tag{C.19}$$

$$= \sum_{i=-\infty}^{+\infty} \sum_{l=-\infty}^{+\infty} \left( \sum_{r=-\infty}^{+\infty} a_{j-r}^\top \overline{a}_{r-l}^\top \right) x_{l-i}^\top \frac{\partial \mathcal{L}}{\partial y_i} \tag{C.20}$$

$$= \sum_{i=-\infty}^{+\infty} \sum_{l=-\infty}^{+\infty} \delta_{j-l} x_{l-i}^\top \frac{\partial \mathcal{L}}{\partial y_i} \tag{C.21}$$

$$= \sum_{i=-\infty}^{+\infty} x_{j-i}^\top \frac{\partial \mathcal{L}}{\partial y_i}, \ \forall j, \tag{C.22}$$

where the second last equality uses Lemma 12. Therefore, we prove $\boldsymbol{a}^\top * \partial \mathcal{L} / \partial \boldsymbol{w} = \boldsymbol{x}^\top * \partial \mathcal{L} / \partial \boldsymbol{y}$, i.e., Equation C.1c in the theorem. Equation C.1b and Equation C.1a can be proved similarly. $\square$

### C.1.2 Proof in Complex Numbers $\mathbb{C}$

In this part, we provide an alternative proof of Theorem 11 using Fourier transform.

*Proof.* If both convolutions in $\boldsymbol{a} * \boldsymbol{y} = \boldsymbol{w} * \boldsymbol{x}$ are circular with period $N$, the celebrated *convolution theorem* relates the discrete Fourier transform of $\boldsymbol{a}$, $\boldsymbol{y}$, $\boldsymbol{w}$ and $\boldsymbol{x}$ with

$$
A_l Y_l = W_l X_l \quad
\begin{cases}
A_l = \displaystyle\sum_{n=0}^{N-1} a_n \omega_N^{nl}, & Y_l = \displaystyle\sum_{n=0}^{N-1} y_n \omega_N^{nl} \\
W_l = \displaystyle\sum_{n=0}^{N-1} w_n \omega_N^{nl}, & X_l = \displaystyle\sum_{n=0}^{N-1} x_n \omega_N^{nl}
\end{cases}
\tag{C.23}
$$

where $\omega_N = \exp(-\mathrm{j}2\pi/N)$ is the $N$-th root of unity. For brevity, we only prove the most difficult equation $-\boldsymbol{a}^\top * \partial\mathcal{L}/\partial\boldsymbol{a} = \boldsymbol{y}^\top * \partial\mathcal{L}/\partial\boldsymbol{y}$ (Equation C.1b) here, and the proofs for the other two equations can be obtained with minor modification.

Taking derivative w.r.t. $A_k$ on both hand sides, we have

$$
\begin{cases}
A_l \dfrac{\partial Y_l}{\partial A_k} = 0, & l \neq k \\
A_l \dfrac{\partial Y_l}{\partial A_k} + Y_k = 0, & l = k
\end{cases}
\tag{C.24}
$$

Since $A_l \neq 0, \forall l$, the equation can be simplified as

$$
\frac{\partial Y_l}{\partial A_k} =
\begin{cases}
0, & l \neq k \\
-\dfrac{Y_k}{A_k}, & l = k
\end{cases}
\tag{C.25}
$$

Then we apply chain rule to obtain the gradient of $A_k$, which yields

$$
\frac{\partial\mathcal{L}}{\partial A_k} = \sum_{l=0}^{N-1} \frac{\partial\mathcal{L}}{\partial Y_l}\frac{\partial Y_l}{\partial A_k} = -\frac{Y_k}{A_k}\frac{\partial\mathcal{L}}{\partial Y_k}.
\tag{C.26}
$$

Again, since $A_k \neq 0, \forall k$, we can simplify the equation as

$$
A_k \frac{\partial\mathcal{L}}{\partial A_k} = -Y_k \frac{\partial\mathcal{L}}{\partial Y_k}.
\tag{C.27}
$$

(Notice that the equation above suggests an efficient algorithm to evaluate the equation using FFT.) To precede, we apply the chain rule one more time to obtain the derivatives w.r.t. $a_n$ and $y_n$ as

$$
\frac{\partial\mathcal{L}}{\partial a_n} = \sum_{k=0}^{N-1} \frac{\partial\mathcal{L}}{\partial A_k}\frac{\partial A_k}{\partial a_n} = \sum_{k=0}^{N-1} \frac{\partial\mathcal{L}}{\partial A_k}\omega_N^{kn},
\tag{C.28a}
$$

$$
\frac{\partial\mathcal{L}}{\partial y_n} = \sum_{k=0}^{N-1} \frac{\partial\mathcal{L}}{\partial Y_k}\frac{\partial Y_k}{\partial y_n} = \sum_{k=0}^{N-1} \frac{\partial\mathcal{L}}{\partial Y_k}\omega_N^{kn}.
\tag{C.28b}
$$

With the equations above, the convolution between $\boldsymbol{a}^\top$ and $\partial\mathcal{L}/\partial\boldsymbol{a}$ can be rewritten as

$$
\sum_{n=0}^{N-1} a_{i-n}^\top \frac{\partial\mathcal{L}}{\partial a_n} = \sum_{n=0}^{N-1} a_{n-i}\frac{\partial\mathcal{L}}{\partial a_{n-i}}
\tag{C.29}
$$

$$
= \sum_{n=0}^{N-1} a_{n-i}\left(\sum_{k=0}^{N-1} \frac{\partial\mathcal{L}}{\partial A_k}\omega_N^{kn}\right)
\tag{C.30}
$$

$$
= \sum_{k=0}^{N-1}\left(\sum_{n=0}^{N-1} a_{n-i}\omega_N^{k(n-i)}\right)\frac{\partial\mathcal{L}}{\partial A_k}\omega_N^{ki}
\tag{C.31}
$$

$$
= \sum_{k=0}^{N-1} A_k \frac{\partial\mathcal{L}}{\partial A_k}\omega_N^{ki}.
\tag{C.32}
$$

With identical arguments, we can rewrite the convolution between $\boldsymbol{y}^\top$ and $\partial\mathcal{L}/\partial\boldsymbol{y}$ as

$$\sum_{n=0}^{N-1} y_{i-n}^\top \frac{\partial\mathcal{L}}{\partial y_n} = \sum_{k=0}^{N-1} Y_k \frac{\partial\mathcal{L}}{\partial Y_k} \omega_N^{ki}. \tag{C.33}$$

Recall the relation in Equation C.27, we have

$$-\sum_{n=0}^{N-1} a_{i-n}^\top \frac{\partial\mathcal{L}}{\partial a_n} = \sum_{n=0}^{N-1} y_{i-n}^\top \frac{\partial\mathcal{L}}{\partial y_n}, \tag{C.34}$$

i.e., $-\boldsymbol{a}^\top * \partial\mathcal{L}/\partial\boldsymbol{a} = \boldsymbol{y}^\top * \partial\mathcal{L}/\partial\boldsymbol{y}$, which completes the proof. $\qquad\square$

## C.2   Efficient Computation using Fast Fourier Transform

The key to speeding up both forward and backward passes in ARMA layers is the *Discrete Fourier Transform* (DFT), along with the *Fast Fourier Transform* (FFT) algorithm.

**Definition 13** (**Discrete Fourier Transform, DFT**). *Given a third-order tensor $\mathcal{T} \in \mathbb{R}^{I_1 \times I_2 \times C}$, we define its DFT of over the spatial coordinates as $\widetilde{\mathcal{T}} \in \mathbb{C}^{I_1 \times I_2 \times C}$.*

$$\widetilde{\mathcal{T}}_{k_1,k_2,c} = \sum_{i_1=0}^{I_1-1} \sum_{i_2=0}^{I_2-1} \mathcal{T}_{i_1,i_2,c}\, \omega_{I_1}^{-i_1 k_1}\, \omega_{I_2}^{-i_2 k_2}, \tag{C.35}$$

*where $\omega_I = \exp(2\pi/I)$ is the $I^{th}$ root of unity. Given the transformed tensor $\widetilde{\mathcal{T}} \in \mathbb{C}^{I_1 \times I_2 \times C}$, the original tensor $\mathcal{T}$ can be recovered by inverse DFT (IDFT) as*

$$\mathcal{T}_{i_1,i_2,c} = \frac{1}{I_1 I_2} \sum_{k_1=0}^{I_1-1} \sum_{k_2=0}^{I_2-1} \widetilde{\mathcal{T}}_{i_1,i_2,c} \omega_{I_1}^{i_1 k_1} \omega_{I_2}^{i_2 k_2}. \tag{C.36}$$

The definition above can be extended to convolutional kernels $\mathcal{A}$ by first zero-padding $\mathcal{A}$ to be $\mathbb{R}^{I_1 \times I_2 \times C}$. With DFT, the autoregressive layer in Equation 5 can be computed as

$$\widetilde{\mathcal{A}}_{k_1,k_2,t}\widetilde{\mathcal{Y}}_{k_1,k_2,t}, = \widetilde{\mathcal{T}}_{k_1,k_2,t}. \tag{C.37}$$

where $\widetilde{\mathcal{A}}, \widetilde{\mathcal{T}}$ are computed from $\mathcal{A}, \mathcal{T}$ with Equation C.35, and $\mathcal{Y}$ is recovered from $\widetilde{\mathcal{Y}}$ by Equation C.36. Similarly, the backpropagation in Equation 4 can be solved as

$$\frac{\partial\mathcal{L}}{\partial\widetilde{\mathcal{A}}_{k_1,k_2,t}} = -\frac{\widetilde{\mathcal{Y}}_{k_1,k_2,t}}{\widetilde{\mathcal{A}}_{k_1,k_2,t}} \cdot \frac{\partial\mathcal{L}}{\partial\widetilde{\mathcal{Y}}_{k_1,k_2,t}}, \tag{C.38a}$$

$$\frac{\partial\mathcal{L}}{\partial\widetilde{\mathcal{A}}_{k_1,k_2,t}} = \frac{1}{\widetilde{\mathcal{A}}_{k_1,k_2,t}} \cdot \frac{\partial\mathcal{L}}{\partial\widetilde{\mathcal{Y}}_{k_1,k_2,t}}. \tag{C.38b}$$

If every DFT is evaluated using FFT, the computational complexity of either forward or backward pass reduces to $O(\log(\max(I_1, I_2))I_1 I_2 T)$, compared to $O((I_1^2 + I_2^2)I_1 I_2 T)$ using Gaussian elimination.

# D   Stability of ARMA Layers

In this section, we will prove the main Theorem 6 in section 5. We organize the section into three subsections: **(1)** In subsection D.1, we formally define the concept of *BIBO stability*, and prove a lemma that relates the stability of a complicated model to the ones of its submodules; **(2)** In subsection D.2, we repeatedly apply the lemma and deduce the stability of an ARMA layer to from the stability of length-3 filters; **(3)** Lastly, in subsection D.3, we prove a theorem on the stability of a length-3 filter.

## D.1   Algebra of BIBO Stability

To analyze the stability of an ARMA model, we adopt the traditional notion of *Bounded-Input Bounded-Output* (BIBO) stability [29] that characterizes the stability of linear systems.

**Definition 14** (**BIBO Stability**). *An input $\boldsymbol{x}$ (or an output ) is bounded if $|x_i| < B_1, \forall i \in \mathbb{Z}$ for some $B_1 > 0$ (or $|y_i < B_2, \forall i \in \mathbb{Z}$ for some $B_2 > 0$). A model is BIBO stable if the output $\boldsymbol{y}$ is bounded given any bounded input $\boldsymbol{x}$, that is*

$$\forall \boldsymbol{x}, \ (\exists B_1 > 0, |x_i| < B_1, \forall i \in \mathbb{Z}) \implies (\exists B_2 > 0, |y_i| < B_2, \forall i \in \mathbb{Z}). \tag{D.1}$$

The following lemma presents that the BIBO stability is preserved under simple algebraic operations of *cascade*, *addition* and *concatenation*. This lemma allows us to reduce the stability analysis of a complex model into its simpler submodules.

**Lemma 15** (**Preserved BIBO Stability**). *BIBO stability is preserved under the operations of cascade, addition, and concatenation. Suppose $f$ and $g$ are two BIBO stable models and consider three compound models: (1) $h_1 = g \circ f$ is a cascaded model $\boldsymbol{y} = h_1(\boldsymbol{x}) = g(f(\boldsymbol{x}))$; (2) $h_2 = f + g$ is a parallel model $\boldsymbol{y} = h_2(\boldsymbol{x}) = f(\boldsymbol{x}) + g(\boldsymbol{x})$; (3) $h_3 = f \otimes g$ is a concatenated model $\boldsymbol{y} = [\boldsymbol{y}^{(1)}, \boldsymbol{y}^{(2)}] = h_3([\boldsymbol{x}^{(1)}, \boldsymbol{x}^{(2)}]) = [f(\boldsymbol{x}^{(1)}), g(\boldsymbol{x}^{(2)})]$. Then $h_1$, $h_2$, and $h_3$ are all BIBO stable.*

*Proof.* (1) *Cascaded model $h_1 = g \circ f$:* $y = h_1(x) = f(g(x))$. Let $t = h(x)$ denote the intermediate result returned by the model $f$. Since $f$ is BIBO stable, we have

$$(\exists B_1 > 0, |x_i| < B_1, \forall i \in \mathbb{Z}) \implies (\exists B_0 > 0, |t_i| < B_0, \forall i \in \mathbb{Z}). \tag{D.2a}$$

Similarly, since $g$ is BIBO stable, we further have

$$(\exists B_0 > 0, |t_i| < B_0, \forall i \in \mathbb{Z}) \implies (\exists B_2 > 0, |y_i| < B_2, \forall i \in \mathbb{Z}). \tag{D.2b}$$

Combining both Equation D.2a and Equation D.2b, we achieve

$$(\exists B_1 > 0, |t_i| < B_1, \forall i \in \mathbb{Z}) \implies (\exists B_2 > 0, |y_i| < B_2, \forall i \in \mathbb{Z}), \tag{D.3}$$

which is the definition of BIBO stability for the model $h_1$.

(2) *Parallel model $h_2 = f + g$:* $\boldsymbol{y} = h_2(\boldsymbol{x}) = f(\boldsymbol{x}) + g(\boldsymbol{x})$. Let $\boldsymbol{u} = f(\boldsymbol{x})$ and $\boldsymbol{v} = g(\boldsymbol{x})$ be the outputs of $f$ and $g$. Since both $f$ and $g$ are BIBO stable, we have the following two relations:

$$(\exists B_1 > 0, |x_i| < B_1, \forall i \in \mathbb{Z}) \implies (\exists B_{21} > 0, |u_i| < B_{21}, \forall i \in \mathbb{Z}), \tag{D.4a}$$

$$(\exists B_1 > 0, |x_i| < B_1, \forall i \in \mathbb{Z}) \implies (\exists B_{22} > 0, |v_i| < B_{22}, \forall i \in \mathbb{Z}). \tag{D.4b}$$

Combining both Equation D.4a and Equation D.4b, we have

$$(\exists B_1 > 0, |t_i| < B_0, \forall i \in \mathbb{Z}) \implies (|y_i| < B_2 = B_{21} + B_{22}, \forall i \in \mathbb{Z}). \tag{D.5}$$

We achieve the definition of BIBO stability for the model $h_2$.

(3) *Concatenated model $\boldsymbol{y} = \boldsymbol{f} \otimes \boldsymbol{g}$:* $\boldsymbol{y} = [\boldsymbol{y}^{(1)}, \boldsymbol{y}^{(2)}] = h([\boldsymbol{x}^{(1)}, \boldsymbol{x}^{(2)}]) = [f(\boldsymbol{x}^{(1)}), g(\boldsymbol{x}^{(2)})]$: Since $f$ and $g$ are both BIBO stable, we have the following relations:

$$(\exists B_1 > 0, |x_i| < B_1, \forall i \in \mathbb{Z}) \implies (\exists B_{21} > 0, |y_i^{(1)}| < B_{21}, \forall i \in \mathbb{Z}), \tag{D.6a}$$

$$(\exists B_1 > 0, |x_i| < B_1, \forall i \in \mathbb{Z}) \implies (\exists B_{22} > 0, |y_i^{(2)}| < B_{22}, \forall i \in \mathbb{Z}). \tag{D.6b}$$

Again, combining both equations we have

$$(\exists B_1 > 0, |x_i| < B_1, \forall i \in \mathbb{Z}) \implies (|y_i^{(2)}| < B_2 = \max(B_{21}, B_{22}), \forall i \in \mathbb{Z}). \tag{D.7}$$

And we achieve the definition of BIBO stability for the model $h_3$. $\qquad \square$

### D.2 Reduction of an ARMA Layer

In what follows, we repeatedly use Lemma 15 to decompose an ARMA layer into simpler systems until the stability analysis is tractable.

**From ARMA Model to AR Model.** In section 4, we show that an ARMA layer can be decomposed into a *cascade* of a *traditional convolutional layer* and an *autoregressive layer* in Equation 5. Since the traditional convolutional layer is always BIBO stable (by triangle inequality), it is sufficient to guarantee the stability of the autoregressive layer:

$$\mathcal{A}_{:,:,t} * \mathcal{Y}_{:,:,t} = \mathcal{T}_{:,:,t}, \ \forall t \in [T]. \tag{D.8}$$

**From Multiple Channels to Single Channel.** Note that the autoregressive layer in Equation D.8 is a *concatenation* of $T$ channels of ARMA models, therefore it is sufficient to guarantee the stability of each ARMA model. For simplicity, we drop the subscript $t$ and denote $\mathcal{A}, \mathcal{Y}, \mathcal{T}$ as $\boldsymbol{A}, \boldsymbol{Y}, \boldsymbol{T}$. Our goal now reduces to finding a sufficient condition for the stability of

$$\boldsymbol{A} * \boldsymbol{Y} = \boldsymbol{T} \iff \sum_{p_1, p_2} A_{p_1, p_2} Y_{i_1 - p_1, i_2 - p_2} = T_{i_1, i_2}, \ \forall i_1, i_2, \tag{D.9}$$

where $\boldsymbol{A} \in \mathbb{R}^{K_a \times K_a}$ and $\boldsymbol{T}, \boldsymbol{Y} \in \mathbb{R}^{I_1 \times I_2}$.

**From Separable 2D-Filter to 1D-Filters.** In a *separable ARMA layer* (Equation 7), each filter $\boldsymbol{A}$ in Equation D.9 is separable, i.e., $\boldsymbol{A} = \boldsymbol{f} \otimes \boldsymbol{g}$ is outer product of two 1D-filters $\boldsymbol{f} \in \mathbb{R}^{I_1}, \boldsymbol{g} \in \mathbb{R}^{I_2}$:

$$A_{p_1, p_2} = f_{p_1} g_{p_2}, \ \forall p_1, p_2. \tag{D.10}$$

Given the factorization, the model in Equation D.9 can be written as a cascade of two submodules:

$$\sum_{p_1} f_{p_1} S_{i_1 - p_1, i_2} = T_{i_1, i_2}, \ \forall i_2, \tag{D.11a}$$

$$\sum_{p_2} g_{p_2} Y_{i_1, i_2 - p_2} = S_{i_1, i_2}, \ \forall i_1, \tag{D.11b}$$

where $\boldsymbol{S} \in \mathbb{R}^{I_1 \times I_2}$ is an intermediate result. Notice that Equation D.11a is a concatenation of $I_2$ submodules, each of which operates on a column of $\boldsymbol{T}$. Similarly, Equation D.11b can be decomposed into a concatenation of $I_1$ submodules, and each submodule operates on a row of $\boldsymbol{S}$. According to Lemma 15, it is sufficient to guarantee the stability of $\boldsymbol{f}$ and $\boldsymbol{g}$ individually. For simplicity, we denote both $\boldsymbol{f}$ and $\boldsymbol{g}$ as $\boldsymbol{a}$, and rewrite each submodule in Equation D.11a or Equation D.11b as

$$\boldsymbol{a} * \boldsymbol{y} = \boldsymbol{x} \iff \sum_{p} a_p y_{i-p} = x_i, \forall i. \tag{D.12}$$

**From General 1D-Filter to Length-3 1D-Filters.** By *the fundamental theorem of algebra*, any one-dimensional filter can be decomposed as a composition of shorter filters [29]. Specifically, suppose $\boldsymbol{a} \in \mathbb{R}^K$ is a filter of length-$K$, it can be factorized into a composition of $Q = (K-1)/2$ length-3 filters such that

$$\boldsymbol{a} = \boldsymbol{a}^{(1)} * \boldsymbol{a}^{(2)} \cdots * \boldsymbol{a}^{(Q)}, \tag{D.13}$$

where each filter $\boldsymbol{a}^{(q)} \in \mathbb{R}^3$ has three coefficients. By the decomposition, the model in Equation D.12 is a cascade of $Q$ submodules

$$\boldsymbol{a}^{(1)} * \left( \boldsymbol{a}^{(2)} * \cdots \left( \boldsymbol{a}^{(Q)} * \boldsymbol{y} \right) \right) = \boldsymbol{x}. \tag{D.14}$$

Therefore, we only need to guarantee the stability for each $\boldsymbol{a}^{(q)}$ individually. In subsection D.3, we will further drop the superscript $q$ and assume $\boldsymbol{a}$ itself is a length-3 filter.

## D.3 Stability of a Length-3 1D-Filter

Without loss of generality, we assume the filter $\boldsymbol{a}$ is centered at 0 with $a_0 = 1$ (otherwise we can rescale the moving-average coefficients). The model at consideration can be written as

$$a_1 y_{i-1} + y_i + a_{-1} y_{i+1} = x_i. \tag{D.15}$$

The stability analysis of this model follows the standard approach using Z-transform [29]. To begin with, we review the concepts of *Z-transform*, *Region of Convergence* (ROC), and their relationships to the BIBO stability of a linear model.

**Definition 16 (Z-transform and ROC).** *Given a one-dimensional sequence $\boldsymbol{h}$, the Z-transform maps the sequence to a complex function on the complex plain $\mathbb{C}$*

$$H(z) = \sum_{i=-\infty}^{+\infty} h_i z^{-i} \tag{D.16}$$

*Notice that the infinite series does not necessarily converge for any $z \in \mathbb{C}$, and the transformation exists only if the summation is convergent. The region in the complex plane that the Z-transform exists is known as the ROC for the sequence $\boldsymbol{h}$.*

**Lemma 17** (**ROC and BIBO Stability**). *Consider a linear model* $\boldsymbol{y} = \boldsymbol{h} * \boldsymbol{x}$*, and let* $H$ *denote the Z-transform of* $\boldsymbol{h}$*, then a necessary and sufficient condition for the model being BIBO stable is that the unit circle belongs to the ROC, i.e., the infinite series*

$$H(e^{\mathrm{j}\omega}) = \sum_{i=-\infty}^{+\infty} h_i e^{-\mathrm{j}\omega i} \tag{D.17}$$

*converges for any frequency* $\omega \in \mathbb{R}$*, i.e.,* discrete-time Fourier transform *(DTFT) exists for* $\boldsymbol{h}$*.*

**Lemma 18** (**ROC of Length-3 AR Model**). *Consider an length-3 AR model* $\boldsymbol{a} * \boldsymbol{y} = \boldsymbol{x}$*, i.e.,* $a_{-1}y_{i-1} + y_i + a_1 y_{i-1} = x_i$*, the Z-transform of  is a length-3 complex polynomial* $A(z) = a_{-1}z + 1 + a_1 z^{-1}$ *with two zeros* $z_1$ *and* $z_2$*. Then the Z-transform of its inverse convolution* $\overline{\boldsymbol{a}}$ *is*

$$\overline{A}(z) = \frac{1}{A(z)} = \frac{z}{a_{-1}z^2 + z + a_1}, \tag{D.18}$$

*with the corresponding ROC* $|z_1| < z < |z_2|$ *as a ring. Since the model can be written as* $\boldsymbol{y} = \overline{\boldsymbol{a}} * \boldsymbol{x}$*, it is BIBO stable if* $|z_1| < 1 < |z_2|$ *according to Lemma 17.*

With the lemmas above, we are ready to prove Theorem 6.

*Proof.* Since the coefficients in $\boldsymbol{a}$ are real numbers, the zeros of $F(z) = zA(z) = a_{-1}z^2 + z + a_1$ are conjugate to each other: **(1)** Both zeros lie on the real axis, i.e., $z_1$ and $z_2$ are real numbers; and **(2)** $z_1$ and $z_2$ are complex conjugate to each other, i.e., $z_1^* = z_2$.

Notice that **(2)** also implies $|z_1| = |z_2|$. However, Lemma 18 shows that $|z_1| < 1 < |z_2|$ is required for BIBO stability, and therefore the second distribution is not feasible.

If both zeros are real, the inequality $|z_1| < 1 < |z_2|$ is equivalent to $F(1) \cdot F(-1) < 0$, i.e.,

$$(a_{-1} + 1 + a_1)(a_{-1} - 1 + a_1) < 0 \tag{D.19}$$
$$\implies (a_{-1} + a_1)^2 - 1 < 0 \tag{D.20}$$
$$\implies |a_{-1} + a_1| < 1, \tag{D.21}$$

which completes the proof. $\qquad\square$

The constrain $|a_{-1} + a_1| < 1$ can be removed by re-parameterizing $(a_{-1}, a_1)$ into $(\alpha, \beta)$:

$$\begin{pmatrix} a_{-1} \\ a_1 \end{pmatrix} = \begin{pmatrix} \sqrt{2}/2 & -\sqrt{2}/2 \\ \sqrt{2}/2 & \sqrt{2}/2 \end{pmatrix} \begin{pmatrix} \alpha \\ \tanh(\beta) \end{pmatrix}, \tag{D.22}$$

where the learnable parameters $(\alpha, \beta)$ are unconstrained. The transformation in re-parameterization is illustrated in the following figure.

**Figure 17:** Visualization of the re-parameterization in Equation D.22.

## Footnotes

[1] `https://challenge2018.isic-archive.com/task1/training/`

[2]`https://github.com/LeeJunHyun/Image_Segmentation`