[Reviews · NeurIPS 2020]

Review 1

Summary and Contributions: The paper tries to resolve a well known problem in current CNN architectures. The strong connection between the depth and receptive field. In order to disentangle the receptive field from the depth they propose a new layer. The basis for the layer is a CNN with an auto-regressive output. They show that this architecture indeed has a larger maximal receptive field and an effective receptive field parametrizable and data dependent. Finally, they show a closed form solution to the gradient estimation problem, an efficient FFT parametrization that allows for only a small overhead with respect to a simple CNN and finally analyse the stability of this layer and suggest simple way to ensure it.

Strengths: The paper contribution is novel and quite interesting. It is potentially interesting both to the applied branch of the field as well as stimulating to future research developments in the domain of architecture design.

Weaknesses: I find the exposition of the FFT not clear enough in the main paper. Also there is little coverage on the consequences of FFT truncation both theoretically and practically. I find the experimental analysis somewhat lacking. Since the theoretical contributions seem to remove most of the impediments to large scale experiments I expected a few standard experiments to "test drive" the approach. I know it is a big effort but I think the method would be quite a bit more compelling if it demonstrates improvements on something like MSCOCO segmentation where object size varies a lot and show if there is benefit there.

Correctness: The paper seems technically correct. I haven't fully checked the ERF and stability line of results.

Clarity: The paper is well written.

Relation to Prior Work: The related work is quite well exposed I find. I would spend just one or two more phrases on the Spatial recurrent neural networks paragraph because that is the closest to the work thus the differences are most important to understand. I would explain why it is not possible to FFT on those but is possible with ARMA. Maybe efficiency implications though that is handled elsewhere.

Reproducibility: Yes

Additional Feedback: - if i understand this correctly the FFT cost is not in the equations in computational overhead. - nit: i think you mean "proceed" not "precede" in your proofs.


Review 2

Summary and Contributions: This paper proposes an autoregressive moving-average (ARMA) layer that can adapt the effective receptive field (ERF) to different tasks and datasets. A practical method to compute the forward and backward pass of the proposed ARMA layer is included. Theoretical analysis on the learning stability is provided as well. The proposed ARMA layer can be used in common networks by replacing the original convolutional layers. Experiments on two datasets (each corresponds to one task) are conducted to evaluate the effectiveness of the ARMA layer.

Strengths: The targeted problem is interesting and important. Enlarging the ERF is a hot topic in deep learning for computer vision. Recent advances like the self-attention mechanism (non-local networks) have been very popular. The introduction of the proposed method is clear and complete. A practical way to compute and train the ARMA layer efficiently is provided, along with a method to increase the learning stability. The proposed method that uses the autoregressive kernel to increase the ERF has some novelty.

Weaknesses: Some claims are not convincing to me. For example, the authors claim in Section 1 that "ARMA networks are complementary to the aforementioned architectures including encoder-decoder structured networks, dilated convolutional networks and non-local attention networks." I agree with the "encoder-decoder structured networks" part but am less convinced that ARMA networks are complementary to dilated convolutional networks and non-local attention networks. In the experimental results in Tables 2&3, using the ARMA layer together with dilated convolutions or the attention mechanism does not bring any improvement. Such results do not suggest the "complementary" relationship. Another example of unconvincing claims lies in Section 7, where the authors claim that the proposed ARMA layer is related to several methods. However, no explanation on how they are related is given. In addition, citations of the mentioned methods are not provided. Besides the above problems, the major weakness of this work is in its experiments, as elaborated below. - Semantic Segmentation on Medical Images: The authors chose the attention u-net [24] as a baseline that uses the attention mechanism. However, [24] actually uses the gate mechanism instead of the attention mechanism in [30,31]. An obvious difference is that the gate mechanism 'filters' each input location by multiplying with a scalar number, while the attention mechanism computes a weighted sum of input locations. In addition, the purpose of using the gate mechanism in [24] is irrelevant to increasing the ERF. In order to compare the ARMA layer with the attention mechanism in the u-net framework, a much more suitable baseline is the AAAI'20 paper "Non-local U-Net for Biomedical Image Segmentation", where non-local attention blocks are used to replace convolutions/deconvolutions in u-net in order to achieve global RF. In addition, since the results in Table 2 are average of 10 runs, std should be reported to make the comparison fair and convincing. - Pixel-level Video Prediction: My major concerns lie in the results in Table 4. Inserting the non-local attention block to Conv-LSTM does not serve as a fair baseline. The functionality of the non-local attention block and the LSTM is overlapping. In [31] and many other studies using the self-attention mechanism, the LSTM is not used at all. Inserting the non-local attention block to Conv-LSTM will not show the true effectiveness of the non-local attention block. The backbone model of this task should be changed to more popular and powerful models, e.g. the backbone model used in [31]. To conclude, 'weak' or inappropriate baselines are used in the experiments, making the results less convincing.

Correctness: The method is technically sound. However, some claims and the experimental designs have flaws, as explained in the 'Weaknesses' section.

Clarity: Yes.

Relation to Prior Work: Yes.

Reproducibility: Yes

Additional Feedback: Comments after author feedbacks: For the semantic segmentation experiments, the comparisons in Feedback Table 1 appropriately support the claims of the authors. However, I'd like to put a remark here. The reported results in Feedback Table 1 look interesting if we compare the numbers with the original Table 2. According to the updated results, the non-local block itself hurts the performance but improves the performance when used together with ARMA layers. There is a possibility that the non-local U-Net baseline was not well tuned. More details of this set of experiments, like model architectures, must be included in the updated version. Meanwhile, I'm not sure whether such "big" revision is allowed in NeurIPS. For the video prediction experiments, I disagree that "the attention mechanism is only added *per-timestep*". Supplementary Figure 8 shows how the authors use the non-local block. It is neither the *per-timestep* nor the original way in [31]. Indeed, I'm NOT expecting the replacement of all the attention mechanisms. However, how the authors use the non-local block doe NOT lead to a fair comparison. Even if ConvLSTM is a good baseline, they should use a 2D non-local block to replace/insert after the original convolutions, in order to compare the effect of expanding the receptive field. I tend to keep my score. I read the other reviewers' comments as well. I sincerely suggest R1 and R4 take a closer look at the experiments.


Review 3

Summary and Contributions: The paper proposes autoregressive moving-average layer (ARMA), a plug-and-play extension of convolution layers that enables the flexible expansion of the CNN models’ effective receptive field. For efficiently performing the forward and backward passes of ARMA, a FFT-based algorithm is proposed. Further techniques are adopted to ensure the training stability. Experimental results show that the integration of ARMA into existing models leads to substantial performance gain in medical image segmentation and video prediction.

Strengths: a) The idea of inserting an additional convolution operation to output neurons is novel, which enables arbitrarily enlarging the receptive field of convolution layers without causing the gridding artifacts that dilated convolution typically have. b) The theoretical soundness of the tunable ERF increase, the FFT-based forward/backward algorithm and the stability constraints of ARMA layers are proved in details. c) Extensive experiments support the effectiveness of the proposed method on different dense prediction tasks. d) The proposed method may be applicable to convolution layers in any CNN model, thus having considerable potential impact in easy improvement over existing models in different problems.

Weaknesses: a) The paper does not suggest strategies to select autoregressive coefficients a under different tasks or other circumstances (e.g. integrating ARMA into multi-scale CNN models) b) It would be more interesting if the paper can discuss the effect of using different autoregressive coefficients for convolution layers at different depths. c) It would be better to also experiment ARMA on regression and generative tasks. After reading the authors' rebuttal, my concerns of the paper have been addressed. I think it's a good paper to be published in ECCV.

Correctness: The claims, the method and the methodology seems correct.

Clarity: The paper is well written. The proposed method and the corresponding proofs are easy to understand. Lucid visualizations are provided to aid the understanding of key concepts.

Relation to Prior Work: Previous works on expanding the CNN effective receptive field are comprehensively discussed, and the main distinctions and contributions of the proposed method are pointed out.

Reproducibility: Yes

Additional Feedback:


Review 4

Summary and Contributions: The paper proposes a new layer for learning dynamically sized effective receptive fields. The layer uses autoregressive transformations after a convolutional layer to transport signals from further away. The paper gives experimental evidence for the effectiveness of this layer.

Strengths: Thorough analysis of the proposed ARMA layer and its computation Proposes an efficient implementation of the ARMA layer that runs in parallel, despite ARMA's sequential definition. Strong on both theorethical and empirical analysis

Weaknesses: Experimental results are not the most convincing Comparison with quite similar methods such as the QRNN [Bradbury et al] missing, although comparison with other methods is mentioned. Would be nice to see evidence on improvement of ARMA also on other tasks with some long range dependencies. Perhaps in Language modelling? Potential technical instability mentioned for ARMA (and how to resolve them). But these could make it harder to adopt or use the layer in practice.

Correctness: Yes. Yes.

Clarity: Pretty well written. Some minor typos.

Relation to Prior Work: Quasi-RNN seem to be the most obvious omission. QRNN also have a convolutional part and a gated autoregressive part to aggregate long-range context. This is similar to ARMA, but contrary to ARMA, QRNN are gated and non-linear.

Reproducibility: Yes

Additional Feedback:

[Author Response · NeurIPS 2020]

**Paper ID: 2499 ARMA Nets.** We thank all reviewers for their valuable feedback, and we are encouraged that all
reviewers found our work important and our method novel. **Major updates of the paper: (1)** To address the concerns
from R3 and R4, we extend Theorem 1 to take into account dilated convolutions and non-uniform layer coefficients.
**Theorem 1 (ERF of a linear ARMA vs CNN network)** *Consider an L-layer linear ARMA network, where the l-th layer computes*
$y^{(\ell)}[i] - a^{(\ell)}y^{(\ell)}[i-1] = \sum_{p=0}^{K^{(\ell)}-1}(1/K^{(\ell)}) \cdot y^{(\ell-1)}[i - d^{(\ell)}p]$. *Suppose* $0 \leq a^{(\ell)} < 1, \forall \ell \in [L]$, *the radius of its ERF is*

$$r(ERF)_{ARMA} = \sqrt{\sum_{\ell=0}^{L-1}\left\{\left[(d^{(\ell)}K^{(\ell)})^2 - 1\right]/12 + a^{(\ell)}/(1-a^{(\ell)})^2\right\}}$$

*When* $a^{(\ell)} = 0, \forall \ell \in [L]$, *the ERF radius of the resulted CNN is* $r(ERF)_{CNN} = \sqrt{\sum_{\ell=0}^{L-1}[(d^{(\ell)}K^{(\ell)})^2 - 1]/12}$.
The result reduces to $\sqrt{L} \cdot \sqrt{(K^2-1)/12 + a/(1-a)^2}$ as in the paper if all layers are identical and the dilation is 1.
**(2)** We change the baseline for semantic segmentation to *"Non-local U-Net for Biomedical Image Segmentation"* (we
add a global aggregating module both at the bottom and up-sampling blocks in U-Net) following R3's suggestion.

Table 1: **Semantic segmentation on ISIC dataset**. For all metrics (ACC, SE, SP, PC, F1 and JS), higher values indicates better performance. The reported numbers are an average of 10 runs with different seeds.

| Model | params. | ACC | SE | SP | PC | F1 | JS |
|---|---|---|---|---|---|---|---|
| Non-local U-Net | 4.403M | $0.945 \pm 0.003$ | $0.877 \pm 0.017$ | $\mathbf{0.973} \pm \mathbf{0.004}$ | $0.844 \pm 0.014$ | $0.831 \pm 0.012$ | $0.741 \pm 0.013$ |
| ARMA-U-Net | 3.455M | $0.955 \pm 0.003$ | $0.896 \pm 0.011$ | $0.972 \pm 0.005$ | $\mathbf{0.873} \pm 0.011$ | $0.861 \pm 0.007$ | $0.780 \pm 0.009$ |
| Non-local ARMA-U-Net | 4.405M | $\mathbf{0.960} \pm 0.002$ | $\mathbf{0.909} \pm 0.009$ | $0.968 \pm 0.004$ | $0.870 \pm 0.011$ | $\mathbf{0.870} \pm 0.006$ | $\mathbf{0.790} \pm 0.008$ |

**R1 - Theoretical and practical analysis of FFT truncation.** The direct application of FFT/DFT assumes the input is
periodically extended (a.k.a. circularly padded). The boundary artifacts by various padding are numerically analyzed in
[r1], and we plan to support reflective padding to mitigate potential artifacts following [r1]. In our applications, we
found such artifacts are imperceptible since the boundary pixels are mostly background.

[r1] Aghdasi F, Ward RK. Reduction of boundary artifacts in image restoration. IEEE TIP. 1996 Apr;5(4):611-8.

**R1 - Large-scale experiments such as MS-COCO segmentation.** We evaluate our approach on an advanced (chal-
lenging) but controllable task of video prediction — our method adapts to predicting objects moving at different speeds,
which requires adaptive ERF. We agree that segmentation with varying object sizes is an interesting future direction.

**R1 - Related works in spatial recurrent neural networks.** Most prior works consider nonlinear RNNs, where the
activation between recursions prohibits an efficient FFT-based algorithm. On the contrary, ARMA is linear between
recursions, thus the recursions reduce to a single deconvolution. We will add this discussion to our related works.

**R3 - Confusing statement "ARMA nets are complementary to the aforementioned architectures..."** By "comple-
mentary", we mean the methods expand the ERF via different ideas. We agree the wording could be confusing and will
modify it. We prove *dilated convolution* is complementary to ARMA in the updated Theorem 1, since dilation ($d^{(\ell)}$)
and auto-regression ($a^{(\ell)}$) contribute to different terms of the ERF. Furthermore, our lightweight ARMA layers can be
used on high-resolution features where *non-local block* could be too expensive in memory and computation.

**R3 - Missing citations in Section 7.** Section 7 is a short conclusion and discussion section; we will add citations to
justify our claims. Specifically, we will add [19, 26] to impulse response filters, [4, 15, 21, 25, 29] to spatial RNNs, and
"Miyato, Takeru, et al. *Spectral normalization for generative adversarial networks.*" to spectral normalization.

**R3 - Inappropriate baseline for semantic segmentation.** We thank the reviewer for suggesting a more appropriate
baseline, and we have updated our experiments (with 10 runs) according to the suggestion. As shown in the updated
Table 1, our ARMA layer is complementary to non-local block — The non-local ARMA U-Net outperforms both
ARMA U-Net and non-local U-Net on most metrics.

**R3 - The choice of our video prediction baseline.** The suggested baseline [31] by the reviewer is for video classifica-
tion instead of pixel-level video prediction. To the best of our knowledge, the state-of-the-art video prediction models
(such as ContextVP [5], PredRNN++ [32]) rely on a ConvLSTM backbone. As shown in Figure 8 (Appendix A.2),
the non-local blocks are inserted between layers, while LSTM mechanism is used between steps: **the LSTM** learns a
long-term dynamic through **time**, while **non-local/ARMA layer** captures large receptive field over **spatial domain**.
Therefore, the functionalities of LSTM and non-local layer are not overlapping, and our baseline is valid and reasonable.

**R4 - Strategies to select autoregressive coefficients.** Since autoregressive coefficients are learnable, we can initialize
them to zeros and let the networks decide the proper coefficients automatically. In Figure 6 (Line 302), we demonstrate
how the networks learn different range of autoregressive coefficients given specific tasks.

**R4 - The effect of using different autoregressive coefficients at different layers.** We update Theorem 1 as in the
beginning of the response — The ERF area (squared radius) is actually a summation of the one for each layer.

**R4 - Experiments on regression and generative tasks.** Video prediction is indeed a dense regression problem, since
a model needs to predict a continuous value for each pixel. Due to space limit, We leave generative task as future work.

[Meta-Review · NeurIPS 2020]

Due to high variance and lack of consensus, an additional expert review was secured, though unfortunately after the rebuttal and discussion phase. While I admit this is not ideal, a clearer picture has now emerged. Most reviewers agree that the proposed model represents a promising direction for learning the receptive field of convolutional networks. It does so in a principled manner based on a novel autoregressive linear layer, and provides a mechanism for stable and efficient training via the FFT. Most reviewers however found the experimental section lacking, either in terms of scale [R1] or baselines [R3, R5]. While large-scale experiments on dense prediction tasks like MSCOCO [R1] or other modalities like text [R5] would have made for a very strong paper, the current choice of benchmarks (medical image segmentation, dense video prediction) seem sufficient, albeit limited. We thank the authors for providing extra results in the rebuttal, which help strengthen the experimental baselines. Despite ongoing concerns from [R3] on baselines, I believe the method will be of great interest to the NeurIPS audience and could prove impactful in improving the performance of CNNs. To that end, I would encourage the authors to release an open-source implementation of the proposed layers. Given the above, I am thus happy to accept the paper for publication, and trust the authors to incorporate reviewer feedback and additional results into the manuscript.